# Probing Predictions on OOD Images via Nearest Categories

**Yao-Yuan Yang***   *yay005@eng.ucsd.edu*
*UC San Diego*

**Cyrus Rashtchian**   *cyroid@google.com*
*Google Research*

**Ruslan Salakhutdinov**   *rsalakhu@cs.cmu.edu*
*Carnegie Mellon University*

**Kamalika Chaudhuri**   *kamalika@eng.ucsd.edu*
*UC San Diego*

**Reviewed on OpenReview:** *https://openreview.net/forum?id=fTNorIvVXG*

## Abstract

We study out-of-distribution (OOD) prediction behavior of neural networks when they classify images from unseen classes or corrupted images. To probe the OOD behavior, we introduce a new measure, *nearest category generalization* (NCG), where we compute the fraction of OOD inputs that are classified with the same label as their nearest neighbor in the training set. Our motivation stems from understanding the prediction patterns of adversarially robust networks, since previous work has identified unexpected consequences of training to be robust to norm-bounded perturbations. We find that robust networks have consistently higher NCG score than natural training, even when the OOD data is much farther away than the robustness radius. This implies that the local regularization of robust training has a significant impact on the network's decision regions. We replicate our findings using many datasets, comparing new and existing training methods. Overall, adversarially robust networks resemble a nearest neighbor classifier when it comes to OOD data[1].

## 1 Introduction

Understanding how neural networks generalize is an ongoing endeavor for machine learning researchers. Many generalization properties for in-distribution data have been discovered (Geirhos et al., 2020; Sagawa et al., 2020; Dasgupta et al., 2022; Chan et al., 2022). However, how deep neural networks generalize on out-of-distribution (OOD) data is less studied. In this context, OOD could mean that the inputs come from previously unseen categories (Salehi et al., 2021; Yang et al., 2021), or that the inputs have been adversarially perturbed (Madry et al., 2017) or corrupted (Hendrycks & Dietterich, 2019). Studying OOD generalization may improve the reliability of machine learning systems. Another goal comes from semi-supervised and self-supervised learning, where the model propagates labels to unlabeled data (Van Engelen & Hoos, 2020). Identifying clear patterns in OOD behavior can aid engineers in choosing among many methods.

Unfortunately, predictions on OOD data can be mysterious. For example, adversarially robust training methods often involve regularizing the network so that it predicts the same label on both a training example and on all points in a small $\epsilon$-ball around the example (Madry et al., 2017; Zhang et al., 2019). Such methods only specify a *local* constraint on prediction behavior. At first glance, it may be tempting to guess that training to be robust in a small $\epsilon$-ball should not affect predictions on other parts of the input space. However, it has become clear that robust training can cause substantial differences in *global* behavior. For

---

*Now at DeepMind.

[1]Code available at https://github.com/yangarbiter/nearest-category-generalization.

example, it may lead to excessive invariances (Jacobsen et al., 2018; Tramèr et al., 2020), improve transfer learning (Salman et al., 2020), or change confidence on OOD data (Hein et al., 2019).

## 1.1 Nearest Category Generalization

In this work, we introduce a new metric that we call Nearest Category Generalization (NCG). Concretely, we calculate how often the predicted label on OOD data matches the 1-nearest-neighbor (1-NN) label and call this measure as the *NCG score* (see Figure 1 for a visual example). We explore whether neural networks are more likely to classify OOD data with the class label of the nearest training input. We also study how adversarial robust training and find that it encourages the network to predict the same label not just in an $\epsilon$-ball but also much further away in the pixel space (Carlini et al., 2019).

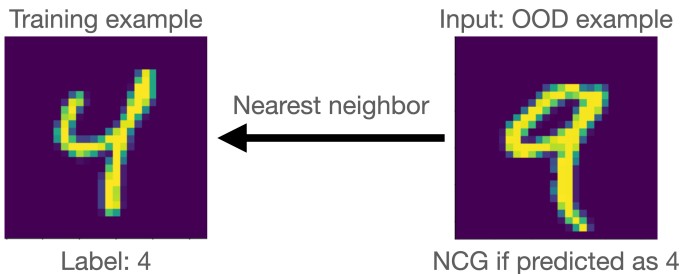

Figure 1: A visual example of nearest category generalization. Assume we have a model trained on MNIST digits zero to eight, and we have images of a nine as OOD examples. In the figure, we see that the input OOD example has the image of a four in the training set as the nearest neighbor. If this OOD example also gets predicted as a four, we say the model is generalizing this OOD example to its nearest category. The *NCG score* measures how often OOD examples are generalized this way on the given model.

Following prior work (Salehi et al., 2021), we consider two canonical types of OOD data (i) **Unseen Classes:** during training, we hold out all images from one of the classes, but during testing, we predict labels for images from this unseen class, (ii) **Corrupted Data:** we train on all classes, but we test on images that have been corrupted. Importantly for us, both sources of OOD data have the property that the distance (e.g., $\ell_2$) is quite large between the OOD data and the standard train/test images. In particular, the distance is much larger than the robustness radius used during robust training, and therefore, the training procedure does not explicitly dictate the predictions on such OOD data. Hence, NCG score measures global resemblance to the 1-NN classifier for unseen or corrupted images.

Understanding NCG can shed new light on many research questions. First, if changes in the training method lead to significant changes in NCG score, then this suggests the network's decision boundaries have shifted to extrapolate very differently on OOD data. Even for ReLU networks, understanding the decision regions far away from training data is an active and important area of research (Arora et al., 2016; Hanin & Rolnick, 2019; Williams et al., 2019). Second, NCG score provides a new way to understand a model's prediction on unlabeled data that cannot be labeled using other information. This is in contrast to transfer learning and few-shot learning that require auxiliary data Raghu et al. (2019); Yosinski et al. (2014); Wang et al. (2020). Overall, we emphasize that the NCG framework is not rooted in a standalone task, but instead, it highlights new prediction patterns of networks on OOD data.

## 1.2 Contributions

Our result shed light on how to examples with unseen classes. Our first finding is that NCG score is consistently higher than chance levels for many training methods and datasets. This means that the behavior on OOD data is far from random. On both natural and corrupted OOD images, the network favors the 1-NN label. Surprisingly, this correlation with 1-NN happens in pixel space with $\ell_2$ distance. The network converges to a classifier that depends heavily on the geometry of the input space, as opposed to making predictions based on semantic or higher-level structures.

Next, we show that robust networks indeed have much higher NCG scores than natural training methods. This holds for a variety of held-out classes, corruption types, and robust training methods. OOD inputs that are classified with the NCG label (1-NN in training set) are considerably further than from their closest training examples. These training examples also have adversarial examples that are closer than the robustness radius $r$ (see Figure 2). This implies that the decision regions of adversarially robust methods extend in certain directions, but not others, and that the local training constraints lead to globally different behavior. We can identify these types of prediction patterns precisely because we examine the network with OOD data.

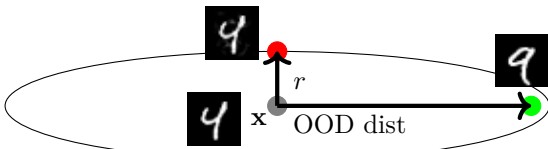

Figure 2: Robust networks tend to predict the same at a large distance in some directions, e.g., toward natural OOD examples (green), but are susceptible to adversarial examples that are closer in the worst-case directions (red).

Besides unseen classes, we also look at corrupted data, such as CIFAR10-C, CIFAR100-C, and ImgNet100-C Hendrycks & Dietterich (2019). The NCG scores for all networks (including natural and robust networks) are above the chance levels, and robust networks often have much higher NCG score than natural training. We also uncover an interesting correlation between NCG score and prediction accuracy for corrupted data. Corrupted examples that are correct in terms of NCG score have a higher chance of being classified correctly in terms of the semantic label as well. In other words, if the network matches the 1-NN prediction, then it is more likely to predict the correct label for a corrupted image. This indicates that having higher NCG score may be a desirable property, as it is positively correlated with better predictions on corrupted data.

Our work uncovers an intriguing OOD generalization property of neural networks, and we find that robust networks have higher NCG score than naturally trained counterparts. However, robust training does not inherently impose constraints on the OOD data that we consider (OOD data are far from perturbed examples). We posit that the NCG behavior is a consequence of the inductive bias produced by neural networks (especially for adversarially robust networks). Also, different forms of robustness ($\ell_2$ and corruptions) may be interconnected with NCG at a deeper level.

### 1.3 Related Work

Many prior works have tried to understand how neural networks generalize. For instance, Geirhos et al. (2020) show that certain types of features are learned more easily, Sagawa et al. (2020) study how changing the architecture could change the features that neural networks learned, and Dasgupta et al. (2022); Chan et al. (2022) study how neural networks generalize on in-distribution data and suggest that in some in-distribution regions, neural networks behave like nearest neighbor classifiers. Our work is a valuable addition to this series of studies.

To investigate the global behavior of robust networks, we explore patterns in how such networks predict on OOD data. However, the dimensionality of the input space for deep neural networks is usually high, making it hard to explore the entire input space. Prior human study shows that when human tends to categorize stimuli that are relatively few and distinct based on other previously seen stimuli that are similar to the current ones (e.g. exemplar-based prediction) (Shepard & Chang, 1963; Nosofsky, 1986; Rouder & Ratcliff, 2004). Dasgupta et al. (2022); Chan et al. (2022) follow this idea and study neural network generalization on in-distribution image and language data. They first train neural networks on data w/ or w/o different attributes (e.g. the existence of a specific word in a sentence). Then, they probe these models' prediction with inputs w/ or w/o these attributes to study the model's generalization property. However, the scope of these prior experiments is limited to in-distribution data or certain OOD data, such as random noise or adversarial perturbations (Hein et al., 2019). Also, in this work, instead of focusing on human interpretable attributes, we focus on the raw input space and the learned feature space.

There has been much research on properties of robust networks beyond robustness. Santurkar et al. (2020) study the performance of adversarially trained models on subpopulation shift. Salman et al. (2020) consider transfer learning for robust models. Kireev et al. (2021) look at accuracy on corrupted data for robust training methods.

Our results on NCG strengthen and complement existing efforts in understanding the excessive invariances that are induced by adversarial training (Jacobsen et al., 2018; Ortiz-Jimenez et al., 2020; Tramèr et al., 2020). Another related area is extrapolation (Balestriero et al., 2021; Xu et al., 2020), where we provide a theoretical result (Theorem 1) that corroborates claims that higher diversity of the training distribution helps extrapolation to linear target functions (Hein et al., 2019; Xu et al., 2020).

Data augmentation is an effective way to improve generalization and robustness (Cubuk et al., 2020; Shorten & Khoshgoftaar, 2019). The success of this approach is consistent with our findings. Local regularization can lead to unexpected behavior, and hence, training on far-away images helps control the decision regions (e.g., for robustness and generalization, cf. Herrmann et al. (2021)).

Connecting NCG with OOD detection is a nice direction for future work (Manevitz & Yousef, 2001; Liang et al., 2018; Ren et al., 2019). OOD detectors already work well for some tasks, such as detecting data from another dataset (Sehwag et al., 2021; Tack et al., 2020). Holding out a single class from the same dataset provides a harder instance of OOD detection. Predictions on unseen data are also studied in the area of open set recognition (Dhamija et al., 2020). Another approach to OOD generalization involves the confidence/uncertainty on OOD data (Kristiadi et al., 2020; Meinke & Hein, 2019; Van Amersfoort et al., 2020). However, much of this work takes a Bayesian perspective and calibrates OOD predictions. We focus on a geometric framework, looking at distances and 1-NN labels in the input space. The perceptual organization of neural networks is another way to probe OOD behavior (Kim et al., 2021).

## 2  Preliminaries

**OOD Data.** We consider two standard types of OOD data. **Unseen Classes:** We hold out all examples from one class during training time (e.g., MNIST without 9s). During test time, we evaluate test accuracy on the remaining classes, and we evaluate NCG score on the held-out class (e.g., we predict 0–8 for the 9s). For each dataset, we hold out different classes, and we use the shorthand `dataset`-wo# to mean that this class # is unseen. For example, MNIST-wo9 is MNIST with unseen digit 9, CIFAR10-wo0 is CIFAR10 with unseen *airplane* and CIFAR100-wo0 is CIFAR100 with unseen *aquatic mammals*. We shorten this as M-9, C10-0, C100-0, etc. We use coarse labels for CIFAR100. For ImageNet, we subsample to 100 classes to form ImgNet100. **Corruptions:** We train on all classes and evaluate standard corruptions, which includes CIFAR10-C and CIFAR100-C (Hendrycks & Dietterich, 2019). Again, we measure NCG score by classifying corrupted data and checking whether the label matches the 1-NN training label.

**Adversarially Robust Training.** Let $\mathcal{B}(\mathbf{x}, r)$ be a ball of radius $r > 0$ around $\mathbf{x}$ in a metric space. A classifier $f$ is said to be *robust* at $\mathbf{x}$ with radius $r$ if for all $\mathbf{x}' \in \mathcal{B}(\mathbf{x}, r)$, we have $f(\mathbf{x}') = f(\mathbf{x})$. Standard adversarially robust training methods such as TRADES (Zhang et al., 2019) work by minimizing a loss function that is the sum of the cross-entropy loss plus a regularization term; this regularization term encourages that the network is smooth in a ball of radius $r$ around each training point $\mathbf{x}_i$, ensuring robustness in this ball. Concretely, the TRADES loss is:

$$\ell(f_\theta(\mathbf{x}_i), y_i) + \beta \max_{\mathbf{x}'_i \in P_i} D_{\mathrm{KL}}(f_\theta(\mathbf{x}'_i), f_\theta(\mathbf{x}_i)), \tag{1}$$

where $\beta$ is a tradeoff parameter, $\ell$ is the cross-entropy loss, and $P_i = \mathcal{B}(\mathbf{x}_i, r)$ is the ball of radius $r$ around $\mathbf{x}_i$.

**Distance Metrics for NCG.** We use the $\ell_2$ distance for two representations. **Pixel Space:** Robust methods aim to have invariant predictions within a small norm ball in pixel space. Hence, we evaluate the distance in pixel space (we believe the $\ell_\infty$ results would be similar). **Feature Space:** For another representation, we first train a different neural network (fully connected MLP) on the in-distribution data (we omit the unseen class). Then we compute the last layer embedding for all images, including those in the unseen class, giving us a learned, latent embedding. This provides vectors for both in-distribution and OOD images, and we use these

vectors as our "feature space" version of the datasets. Note that we *do not* claim that these representations capture human-level or semantic similarity for the OOD data. Nonetheless, they both can be used by future work to provide insight into the OOD prediction behavior of robust and normal networks.

**NCG score.** The NCG score is the fraction of OOD inputs that are labeled as their nearest neighbor in the training set (i.e., we measure agreement with the 1-NN classifier). For corrupted data, we use the whole training set. For unseen classes, we use the training set minus the held-out class. We measure the 1-NN prediction in $\ell_2$ distance in either the pixel space or the above-defined feature space.

## 3   NCG for Unseen Classes

**Set-up.** For natural/robust training, on MNIST, we evaluate a CNN; on CIFAR10/100, we use Wider ResNet (WRN-40-10); on ImageNet, we use ResNet50. We consider natural training and mixup (Zhang et al., 2017) as baselines. For robust training, we consider two standard methods, Adversarial Training (Madry et al., 2017)(AT) and TRADES (Zhang et al., 2019). These methods are known to have high adversarial robustness to $\ell_2$ perturbations, and hence, they serve as good baseline examples of robust networks. For TRADES, we use robustness radii $r \in \{2, 4, 8\}$ for the $\ell_2$ ball in Equation (1). In the pixel space, we use $r = 2$ for AT. In the feature space, we set $r = 1$ for AT on CIFAR10/100, and $r = .5$ for AT on ImgNet100 (on CIFAR10/100, AT failed to converge with $r = 2$ and on ImgNet100 with $r \in \{1, 2\}$). We denote TRADES with $r = 2$ and AT with $r = 1$ as TRADES(2) and AT(1), respectively. Prior work observes that AT and TRADES give similar results with parameter tuning (Yang et al., 2020; Carmon et al., 2019), and hence we expect them to behave similarly. Appendix C has more details.

**Datasets.** We consider all 10 classes as the unseen class for MNIST and three unseen classes for each of CIFAR10, CIFAR100, and ImgNet100. CIFAR10, we consider removing the *airplane*, *deer*, and *truck* classes; for CIFAR100, we remove the *aquatic mammals*, *fruit and vegetables*, and *large man-made outdoor things* classes; for ImgNet100, we remove the *American robin*, *Gila monster*, and *eastern hog-nosed snake* classes. These are denoted as CIFAR10-wo{0, 4, 9}, CIFAR100-wo{0, 4, 9}, ImgNet100-wo{0, 1, 2}.

**Results.** Table 1 shows the NCG score of natural and robust models averaged over unseen classes (in both pixel and feature space). We perform a chi-squared test against the null hypothesis that the distribution of the labels is uniform with the $p$-value threshold set to 0.01. We find that for **all** 80 models trained, there is a significantly higher than chance level NCG score. Then, we perform a $t$-test between each robust model vs. natural training, with the null hypothesis being that the robust model has lower NCG score. In Table 2, we show the number of cases that pass this $t$-test. Adversarially robust models, TRADES and AT, almost always have a higher NCG score than natural training with verified statistical significance. Meanwhile, mixup have a higher NCG score than natural training in only less than half of the cases.

**Adversarial robustness increases NCG score.** The unseen class is absent at training, and this property has been obtained simply by making the model adversarially robust. This is interesting because it suggests that robust models extrapolate to OOD data in a way that is more likely to match 1-NN predictions. Prior work on extrapolation (Xu et al., 2020) has shown that MLPs tend to extrapolate as linear functions on far OOD data. The 1-NN classifier is not a linear function in their sense. Specifically, the 1-NN label is determined by the Voronoi decomposition of the training data, and the decision boundaries separate far away points using hyperplanes. Hence, the higher NCG score of robust classifiers uncovers a new phenomenon of OOD behavior. We next investigate whether we should expect higher NCG score for robust models by measuring the distance to OOD examples.

### 3.1   OOD Data are Farther than Adversarial Examples

Why do robust models have a higher NCG score for unseen classes? One plausible explanation is that the robust methods enforce the neural network to be locally smooth in a ball of radius $r$; if the OOD inputs are closer than $r$ from their nearest training example, then they would get classified accordingly. Next, we test if this is the case by measuring the $\ell_2$ distance between each OOD input $\mathbf{x}$ and its closest training example $\tilde{\mathbf{x}}$. Then, we calculate the closest adversarial example $\mathbf{x}'$ to $\tilde{\mathbf{x}}$ using various attack algorithms. We measure (i) the distance between OOD example and its closest training example ($\|\mathbf{x} - \tilde{\mathbf{x}}\|_2$) and (ii) the empirical

Table 1: The average and standard deviation of the NCG scores across different held-out class of each dataset. There are 10 unseen classes for MNIST and 3 unseen classes for CIFAR10, CIFAR100, and ImgNet100. In general robust methods have a higher average NCG than natural training. The chance level is $1/9$ for MNIST and CIFAR10, $1/19$ for CIFAR100, and $1/99$ for ImgNet100.

|  | MINST | CIFAR10 | CIFAR100 | ImgNet100 |
|---|---|---|---|---|
|  | pixel | | | |
| natural | $.49 \pm .14$ | $.24 \pm .09$ | $.18 \pm .03$ | $.04 \pm .01$ |
| mixup | $.47 \pm .14$ | $.23 \pm .09$ | $.20 \pm .06$ | $.04 \pm .01$ |
| TRADES(2) | $.59 \pm .12$ | $.34 \pm .12$ | $.29 \pm .09$ | $.04 \pm .01$ |
| TRADES(4) | $.59 \pm .10$ | $.37 \pm .11$ | $.29 \pm .10$ | $.05 \pm .01$ |
| TRADES(8) | $.52 \pm .12$ | $.34 \pm .10$ | $.29 \pm .13$ | $.06 \pm .01$ |
| AT(2) | $.60 \pm .10$ | $.36 \pm .12$ | $.26 \pm .07$ | $.04 \pm .01$ |
|  | feature | | | |
| natural | $.50 \pm .18$ | $.82 \pm .02$ | $.66 \pm .03$ | $.12 \pm .01$ |
| mixup | $.56 \pm .16$ | $.79 \pm .02$ | $.66 \pm .03$ | $.14 \pm .01$ |
| TRADES(2) | $.58 \pm .15$ | $.82 \pm .01$ | $.72 \pm .02$ | $.16 \pm .01$ |
| TRADES(4) | $.64 \pm .10$ | $.85 \pm .02$ | $.71 \pm .02$ | $.13 \pm .01$ |
| TRADES(8) | $.67 \pm .11$ | $.85 \pm .02$ | $.71 \pm .02$ | $.14 \pm .01$ |
| AT(2)/(1)/(.5) | $.54 \pm .17$ | $.85 \pm .03$ | $.73 \pm .02$ | $.16 \pm .01$ |

Table 2: The number of robust models with higher NCG score than natural training. For MNIST we check 10 unseen classes, and for CIFAR10, CIFAR100, and ImgNet100, we use 3 unseen classes. 10/10 means that out of the 10 unseen classes, all 10 models have a higher NCG score.

|  | pixel | | | | feature | | | |
|---|---|---|---|---|---|---|---|---|
|  | M | C10 | C100 | I | M | C10 | C100 | I |
| mixup | 5/10 | 0/3 | 2/3 | 0/3 | 0/10 | 0/3 | 2/3 | 3/3 |
| TRADES(2) | 10/10 | 3/3 | 3/3 | 3/3 | 9/10 | 2/3 | 3/3 | 3/3 |
| TRADES(4) | 8/10 | 3/3 | 3/3 | 3/3 | 10/10 | 3/3 | 3/3 | 3/3 |
| TRADES(8) | 7/10 | 3/3 | 3/3 | 3/3 | 10/10 | 3/3 | 3/3 | 3/3 |
| AT(2)/(1)/(.5) | 10/10 | 3/3 | 3/3 | 3/3 | 9/10 | 3/3 | 3/3 | 3/3 |

robustness radius ($\|\mathbf{x}' - \tilde{\mathbf{x}}\|_2$). We plot the histograms in Figure 3. We use the C&W attack Carlini & Wagner (2017) algorithm to find the closest adversarial example. Some of these attacks are slow, so we compute adversarial examples for 300 randomly sampled training examples that are correctly predicted. Then, we restrict to OOD examples that have one of these 300 training examples as their closest neighbor.

Figure 3a reports a typical distance histogram in the pixel space (for C10-0). We find that the histograms of OOD distances and the empirical robust radii have little to no overlap in the pixel space, while in the feature space, there is some overlap but not much. To better understand what is happening, we measure the percentage of OOD examples that are covered in the ball centered around the closest training example with a radius of the empirical robust radius. We find that in both the pixel and feature space, for 186 out of 190 models, this percentage is less than 2%, which is significantly smaller than the difference between the NCG score of robust and naturally trained models in most cases (190 comes from having two metric spaces, five models, and 19 datasets). This result shows that almost all OOD examples from unseen classes are significantly further away from their closest training example than the empirical robust radius of these training examples.

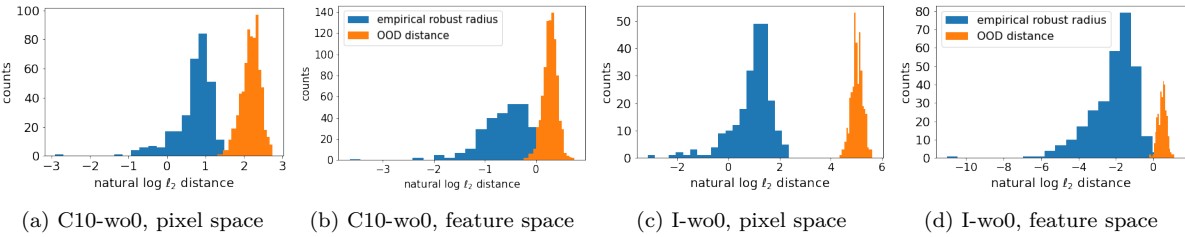

(a) C10-wo0, pixel space    (b) C10-wo0, feature space    (c) I-wo0, pixel space    (d) I-wo0, feature space

Figure 3: Histograms: log of the empirical robust radius and OOD distance for TRADES(2) on CIFAR10-wo0 and ImgNet100-wo0. Adversarial examples are much closer than OOD examples.

## 3.2  The Role of the Training Procedure

We next ask whether changing the robustness regions $P_i$ when optimizing the loss in Equation (1) can change NCG score. TRADES enforces smoothness in a region $P_i$ that is to be a fixed radius $r$ norm ball around each training example. Enforcing smoothness on fixed radius norm balls may not ensure good NCG score. Figure 4c shows an example – here, the purple points are closer to the orange cluster on the left and further from the orange cluster on the right. If we only enforce smoothness on a uniform ball (TRADES), the decision boundary for the purple points does not extend right enough. We explore making the classifier smooth in regions $P_i$ that *adapt to the geometry of the dataset.*

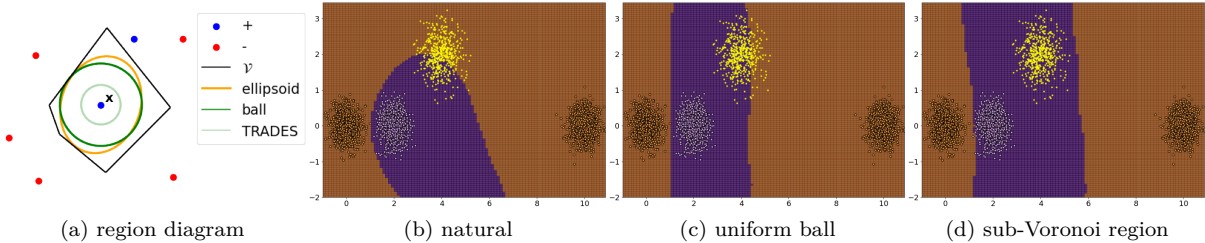

(a) region diagram    (b) natural    (c) uniform ball    (d) sub-Voronoi region

Figure 4: (a) A diagram showing the difference between the sub-Voronoi region ($\mathcal{V}$) and the ball $\mathcal{B}$ used to approximate it. In figure (b), (c), and (d), we plot the decision boundary of neural networks trained with natural training, TRADES, and enforcement on the smoothness in $\mathcal{V}$. The yellow examples are the OOD examples, and they are closer to the purple examples. In (b) and (c), we see that the predictions on the yellow examples are not consistent to the nearest neighbor; on the other hand, in (d), the yellow examples are predicted as purple.

**Sub-Voronoi Region.** Can we use the 1-NN classifier as a guide to actively improve NCG score? While we do not see OOD inputs at training, we can encourage all points $\mathbf{x}$ that are closer to a training point $(\mathbf{x}_i, y_i)$ than to any other training point with a different label to be assigned to label $y_i$. In other words, we could set $P_i$ to be the Voronoi region of $\mathbf{x}_i$ in the union of $(\mathbf{x}_i, y_i)$ and all other training points whose labels are different from $y_i$. We call this the *sub-Voronoi region* of $(\mathbf{x}_i, y_i)$. Figure 4a shows an example.

Figures 4b to 4d show how different training methods change the decision regions. Note that the yellow points are never seen during training or testing (they are just for illustration of hypothetical OOD inputs). We draw the decision boundary for three methods – natural, TRADES, and training with $P_i$ set to the sub-Voronoi region in Equation (1). All three perform well on in-distribution examples; however, unlike TRADES, optimizing over the sub-Voronoi region classifies all of the OOD examples with their nearest categories by putting the purple boundary in the correct location.

We note that in Figure 4, we do not claim that one decision boundary is better than the others. Our motivation is to see how to increase NCG score through a new training method. NCG score measures how well the decision boundary of a neural network matches the decision boundary of a 1-nearest neighbor classifier

on OOD examples. Thus, the sub-Voronoi regions formed by the training examples are a natural choice of the robust region (because if a classifier has a decision boundary that matches the sub-Voronoi region, and the label of each region is the same as the training example in that region, then the classifier has a 100% NCG score). The reason for examining this property is that if an OOD example is given with no assumption made on these given examples. Then the best thing we can do is to predict this given example with the same label as the closest training example.

### 3.3 Approximations to the sub-Voronoi region

The loss in Equation (1) is minimized by an iterative procedure. At each iteration, we find the input in $P_i$ that maximizes the regularization term. Running this requires being able to project onto $P_i$ efficiently. While this can be done relatively fast when $P_i$ is a constant radius ball, it is considerably more challenging for the sub-Voronoi region – which is a polytope with close to $n$ constraints (one for each training point with label $\neq y_i$). Therefore, we consider three alternative approximations that are faster to project on and can be efficiently implemented during training.

**Sub-sampled sub-Voronoi.** Here, instead of the sub-Voronoi region, of $\mathbf{x}_i$ in the full training data, we use the Voronoi cell of $\mathbf{x}_i$ in the union of $(\mathbf{x}_i, y_i)$ and a subsample of the training data with labels not equal to $y_i$. Since the sub-sampled sub-Voronoi region can be large, which can cause the network to underfit, we introduce a shrinkage parameter $\lambda \in [0, 1]$ to scale down the size of the region.

**Ellipsoid.** An alternative method is to use an ellipsoidal approximation. Computing the maximum volume ellipsoid inside the region is again challenging. To improve efficiency, we use the following approximation. We pick $k$ differently-labeled training examples that are closest to $\mathbf{x}_i$, learn a PCA on these $k$ examples, and pick the ellipsoid centered at $\mathbf{x}_i$ and described by the top $k/2$ principal components. We use a shrinkage parameter $\lambda$ to help generalization.

**Non-uniform ball.** A final approximation is to use an $\ell_2$ ball of radius $r_i$, where $r_i$ is set to half the distance between $\mathbf{x}_i$ and its closest training point with a different label; this is the largest ball centered at $\mathbf{x}_i$ that is contained in the sub-Voronoi region. As with the previous methods, with finite training data, this may overestimate the region where we should predict $y_i$ and hence lead to underfitting; to address this, we again introduce a shrinkage parameter $\lambda$, setting $P_i$ to $\mathcal{B}(\mathbf{x}_i, \lambda r_i)$. More details about the minimization procedure and each of these alternatives are given in Appendix B.

### 3.4 Experiments on how the robust region affects NCG

We now empirically measure how the role of changing $P_i$ affects NCG score on real data. For this purpose, we consider enforcing smoothness in the three types of regions discussed above – non-uniform ball, ellipsoid, and sub-sampled sub-Voronoi. We also look at TRADES with three different radii and natural training as baselines. A detailed discussion of the experimental setup is in Appendix C. Note that we do not aim to achieve the best performance in any given measure, so we mostly use standard parameter settings, and we do not use data augmentation.

**Results and Discussion.** Table 3 shows the train, test, NCG score for MNIST, CIFAR10, CIFAR100 with different unseen categories. All robust methods – TRADES and three approximations to sub-Voronoi – have higher NCG score than the natural training. This agrees with our previous observations. Surprisingly, there is a lot of variation in the results. This warrants investigation, but we do not have a clear conjecture as to why certain classes have higher NCG score. For instance, in M-4, the NCG score can be up to 83%, while in M-1, the best is 53%. Perhaps there is a visual similarity between some classes of images (e.g., 4s and 9s look alike), or spurious correlations (Veitch et al., 2021), or only some datasets have sufficient diversity in examples to "cover" the OOD class (Hein et al., 2019; Xu et al., 2020). In high dimensional space, it is hard to measure whether some of the unseen classes are more like the training data than others. We later explore the NCG score as a function of the distance to the training set. At least in pixel space and for color images, we find that closer images are more likely to receive the NCG label. We next consider corrupted images as another source of OOD data.

Table 3: The training, testing and NCG score of networks trained by enforcing smoothness on different regions (pixel space). We use MNIST with digits 0, 1, 4, and 9 as the unseen classes, CIFAR10 with *airplane* and *deer* as the unseen classes, and CIFAR100 with *aquatic mammals* and *fruit and vegetables* as the unseen classes.

| | trn acc. | tst acc. | NCG score | trn acc. | tst acc. | NCG score |
|---|---|---|---|---|---|---|
| | | MNIST-wo0 (M-0) | | | MNIST-wo1 (M-1) | |
| sub-voronoi | 0.981 | 0.981 | 0.474 | 0.982 | 0.981 | 0.376 |
| ellipsoid | 0.981 | 0.981 | 0.476 | 0.982 | 0.980 | 0.425 |
| ball | 0.975 | 0.973 | **0.510** | 0.976 | 0.973 | 0.338 |
| TRADES | 0.954 | 0.956 | 0.485 | 0.975 | 0.974 | **0.528** |
| nat | 1.000 | 0.995 | 0.390 | 1.000 | 0.995 | 0.273 |
| | | MNIST-wo4 (M-4) | | | MNIST-wo9 (M-9) | |
| sub-voronoi | 0.982 | 0.983 | **0.827** | 0.988 | 0.988 | 0.703 |
| ellipsoid | 0.982 | 0.982 | 0.820 | 0.988 | 0.988 | **0.725** |
| ball | 0.977 | 0.976 | 0.795 | 0.982 | 0.981 | 0.711 |
| TRADES | 0.988 | 0.987 | 0.810 | 0.962 | 0.964 | 0.703 |
| nat | 1.000 | 0.995 | 0.760 | 1.000 | 0.996 | 0.577 |
| | | CIFAR10-wo0 (C10-0) | | | CIFAR10-wo4 (C10-4) | |
| sub-voronoi | 0.735 | 0.658 | 0.452 | 0.486 | 0.482 | **0.417** |
| ellipsoid | 0.671 | 0.613 | 0.472 | 0.483 | 0.481 | 0.409 |
| ball | 0.794 | 0.618 | **0.530** | 0.871 | 0.664 | 0.317 |
| TRADES | 0.870 | 0.660 | 0.520 | 0.862 | 0.643 | 0.355 |
| nat | 1.000 | 0.900 | 0.362 | 1.000 | 0.886 | 0.222 |
| | | CIFAR100-wo0 (C100-0) | | | CIFAR100-wo4 (C100-4) | |
| sub-voronoi | 0.308 | 0.289 | 0.241 | 0.706 | 0.499 | **0.207** |
| ellipsoid | 0.633 | 0.478 | 0.255 | 0.466 | 0.385 | 0.198 |
| ball | 0.936 | 0.517 | 0.236 | 0.930 | 0.489 | 0.176 |
| TRADES | 0.891 | 0.534 | **0.264** | 0.995 | 0.534 | 0.193 |
| nat | 1.000 | 0.757 | 0.169 | 1.000 | 0.694 | 0.140 |

## 4 NCG for Corrupted Data

Do the trends that we have seen for NCG also hold for other OOD data besides unseen classes? We consider images with Gaussian noise, blur, JPEG artifacts, snow, speckle, etc (Hendrycks & Dietterich, 2019). We use "-C" to denoted the corrupted version of a dataset, e.g., CIFAR10-C (C10-C), CIFAR100-C (C100-C), and ImgNet100-C (I-C), which consists of corrupted images from the CIFAR10 (C10), CIFAR100 (C100), and ImgNet100 (I) datasets. C10-C and C100-C have 18 kinds of corruption, each with 5 corruption levels. I-C has 15 kinds of corruption, each with 5 levels. We consider models trained on regular datasets, C10, C100, and I (instead of removing the unseen class). We use *corrupted set* to refer to a corruption type and intensity level. For C10 and C100, there are 90 corrupted sets for each dataset; for I, there are 75 corrupted sets. We consider NCG under $\ell_2$ distance for both the pixel and learned feature spaces. We train on C10, C100, and I and measure NCG score on C10-C, C100-C, and I-C, respectively; each training method is measured on 255 corruption sets.

**Results.** In both pixel and feature space, we find that **all** the 255 corruption sets have an NCG score above chance level. For robust models, we find that in the pixel space, TRADES(2) has an NCG score higher than naturally trained models on **all** 255 corrupted sets. Quantitatively, on average (over the 90 and 75 corrupted sets), TRADES has an NCG score that is $1.35 \pm .02$, $1.36 \pm .03$, and $1.66 \pm .04$ times higher than naturally trained models for CIFAR10, CIFAR100, and ImgNet100 respectively. Hence, in pixel space, the TRADES

training procedure leads to much higher NCG score for corrupted data. On the other hand, in the feature space, the results are much less conclusive. In particular, the robust models still have higher NCG score on average, but this does not happen consistently across corruption types or datasets (see Appendix D.7).

**Discussion.** Our findings in Section 3 extend to these corruptions as the OOD data. In particular, the NCG score of adversarially robust networks is higher on both unseen, natural images, and on corrupted versions of seen classes. On the other hand, in the feature space, the robust models do not have much difference in NCG score from the naturally trained models. The fact that we see less variation in the feature space compared to the pixel space is consistent with our results on the unseen classes. The TRADES robust training does not affect the decision regions as much when the smoothness is enforced in the feature space. This is likely because the learned embedding has less variation in elements of the same class, and hence, the TRADES training does not contribute as much.

### 4.1 NCG score vs. test accuracy

Next, we look at the interaction between the NCG and test accuracies, so we also measure the test accuracy on the NCG correct data and NCG incorrect data. We first observe that NCG correct examples are more likely to be correctly classified. To verify that this phenomenon is statistically significant across the board, we perform the one-sided Welch's t-test (which does not assume equal variance) with the null hypothesis being that the accuracy of NCG correct example is not greater than the accuracy of NCG incorrect example. We set the p-value threshold to 0.05, and the test results are in Table 4. For more details, please refer to Appendix D.3.

Table 4: Number of cases where the NCG correct examples have a **significantly** higher test accuracy than the NCG incorrect examples. 87/90 means that out of the 90 corrupted sets, 87 of them pass the t-test.

|            | pixel | | | feature | | |
|------------|-------|-------|-------|-------|-------|-------|
|            | C10   | C100  | I     | C10   | C100  | I     |
| natural    | 87/90 | 87/90 | 57/75 | 88/90 | 90/90 | 73/75 |
| TRADES(2)  | 84/90 | 88/90 | 60/75 | 89/90 | 90/90 | 73/75 |

## 5  Discussion and Connections

### 5.1  Sample complexity of NCG vs. OOD detection

We first observe that NCG is an easier problem in theory than OOD detection. Indeed, any time we can detect an OOD example, then we can use the 1-NN classifier to label it. Thus, the NCG score should always be at least as high as the true positive rate for OOD detection.

We next theoretically show that the converse is false in general. We prove that there exist cases where maximizing NCG score can be significantly more sample efficient than solving detection problems. OOD detection is hard when certain regions of space have low mass under the input distribution. In this case, it takes many samples to see a representative covering of the support. Prior work has made similar high-level observations in the context of data diversity for robustness and extrapolation (Hein et al., 2019; Xu et al., 2020). In contrast, when the NCG label is the same for nearby regions, then it suffices to see samples from fewer regions and generalize accordingly. We formalize this claim in the following theorem, which identifies simple distributions where detection requires many more samples than NCG. While the proof is not complicated, our result suggests that achieving a high NCG score can be much easier than achieving a good detection rate in some case

**Theorem 1** *For any $\epsilon \in (0, 1/2)$, $d \geqslant 1$, and $C \geqslant 2$, there exists distributions $\mu$ on training examples from $C$ classes in $\mathbb{R}^d$ and $\nu$ on OOD test examples from outside of $\mathsf{supp}(\mu)$ such that (i) detecting whether an example is from $\mu$ or $\nu$ requires $\Omega(C/\epsilon)$ samples from $\mu$, while (ii) classifying examples from $\nu$ with their nearest neighbor label from the support of $\mu$ requires only $O(C \log C)$ samples.*

Figure 5 shows intuition for Theorem 1 in the binary case. OOD examples come from outside of the colored cubes. Appendix A has the proof for $C$ classes in $\mathbb{R}^d$.

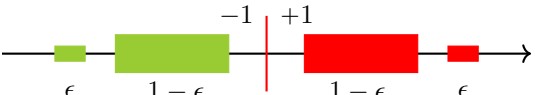

Figure 5: An example for Theorem 1. The sample frequency is the size of the red/green shapes. With a few samples from each large probability region, we determine the NCG label via a large margin solution, but OOD detection requires samples from the small probability regions.

We sketch how to generalize Figure 5 for more classes and higher dimensional data. The idea is that we translate and replicate the binary dataset and increase the regions to $d$-dimensional cubes. For the distribution $\mu$, we have $4C$ cubes with side length $1/\sqrt{d}$. There will be 2 cubes that have labels from each of the $C$ classes. The high probability cubes emit samples with probability $\approx (1-\epsilon)/C$ and the lower probability with $\approx \epsilon/C$. Due to the side lengths being $1/\sqrt{d}$, the 1-nearest neighbor (1-NN) in $\ell_2$ of the low probability region is paired with an adjacent high probability box, and hence, it is easy to predict given samples from the high probability region. By a coupon collector argument, we see all high probability regions after $O(C \log C)$ samples. On the other hand, by the construction of the probability distributions, we need $\Omega(C/\epsilon)$ samples for OOD detection, where $\epsilon$ is the sample probability from a low probability region. For the OOD distribution $\nu$, we strategically sample points from outside of all of these cubes (while guaranteeing that the nearest neighbor labels are still correct). Thus, $O(C \log C)$ samples are sufficient for NCG, but $\Omega(C/\epsilon)$ are needed for OOD detection.

This result suggests that there are cases where OOD detection is extremely sample inefficient and cannot be done well. In these cases, the model will have to give a prediction on examples that it does not expect. Thus, it would be crucial to understand how the model predicts these examples.

As a concrete example, we train a model on MNIST images of 0-8 and use the model for prediction on images of 9s. We also train an OOD detector – ODIN (Liang et al., 2018) – which has a .951 true positive rate and .875 false negative rate. In this example, many images of 9s cannot be easily picked out by OOD detectors and will be treated as in-distribution examples. Thus, it is important to know what kind of prediction will be given to these 9s. From our previous result, we find that many 9s are predicted as 4s, and this can be explained by nearest category generalization.

### 5.2  When do we have higher NCG scores?

One hypothesis is that OOD examples that are further from the training set are less likely to be predicted with the NCG label. To check this hypothesis, we conduct the following experiment. We bin the OOD examples based on their distance to the closest training example into 5 equal size bins, and we evaluate the NCG score in each bin. A typical result is shown in Figure 6 (additional results are in Appendix D.6.3). We find that the NCG score is generally higher when OOD examples are closer to the training examples. This is true both for an unseen class and for corrupted data (in aggregate). While this is not surprising, it does give more insight into the patterns of OOD data that are labeled with the 1-NN label. We also looked at MNIST, but there was no clear connection between distance and NCG score.

It is known that OOD detectors perform well when in- and out-of-distribution data are far away from each other (Liang et al., 2018). This, along with our result, gives us an interesting dynamic, which is that neural networks behave more like the nearest neighbor classifier when detectors perform worse. This also suggests that many of the OOD examples that are misclassified as in-distribution examples could follow the NCG property. It would allow the user to know that even when the OOD detector fails, the model would still output something reasonable. Therefore, if one wants a robust and predictable prediction on OOD data, it can be desirable to have a high NCG score.

**Limitation: Choice of the distance metric.** We only evaluate $\ell_2$ distance in the pixel and feature space. With $\ell_2$ distance, we already discovered interesting OOD behavior. However, an important direction is to explore other distance measures to understand the prediction patterns. Some options include $\ell_\infty$ or cosine

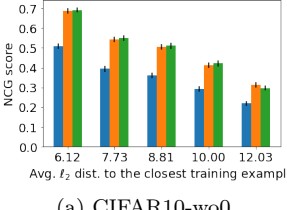 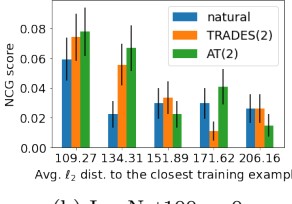 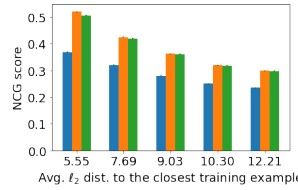 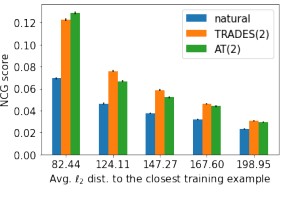

| (a) CIFAR10-wo0 | (b) ImgNet100-wo0 | (c) CIFAR10 (corrupted) | (d) ImgNet100 (corrupted) |

Figure 6: We group OOD examples into five bins based on $\ell_2$ distance to the closest training example in pixel space. (a) and (b) show the NCG score of each bin on the unseen class of CIFAR10-wo0 and ImgNet100-wo0. (c) and (d) the NCG score of each bin on the aggregate corrupted data of CIFAR10 and ImgNet100. The downward trend seen here is not as apparent in the feature space (see Appendix D.6.3).

distance or measuring distance in a embedding space of an auto-encoder. Ideally, the distance measure would capture perceptual similarity of the images. This would imply that NCG score corresponds to how humans may predict an unseen class. However, it is not clear if such a perceptual metric exists for images.

## 6 Conclusion

We examine out-of-distribution (OOD) properties of neural networks and uncover intriguing generalization properties. Neural networks have a tendency of predicting OOD examples with the labels of their closest training examples. We measure this via a new metric called NCG score. Robust networks consistently have higher NCG score than naturally trained models. We replicate this result for two sources of OOD data (unseen classes and corrupted images), and we experiment with a variety of new and existing robust training methods. This is surprising because the OOD data are much further away in $\ell_2$ distance than both the robustness radius and the nearest adversarial examples. Therefore, the robust training procedure is changing the decision regions on parts of space that are not directly considered in the loss function. We posit that this behavior and the higher NCG score is a consequence of the inductive bias of robust networks. In the future, it would be interesting to evaluate NCG for other training methods, architectures, and sources of OOD data like distribution shift or spurious correlations. Overall, NCG can be a valuable and scalable addition to the toolbox of evaluation metrics for OOD generalization.

### Acknowledgments

We thank Angel Hsing-Chi Hwang, Hanie Sedghi, Mary Anne Smart, and Otilia Stretcu for providing thoughtful comments on the paper. Kamalika Chaudhuri and Yao-Yuan Yang thank NSF under CIF 1719133, NSF under CNS 1804829, and ARO MURI W911NF2110317 for support. This work was also supported in part by NSF IIS1763562 and ONR Grant N000141812861.

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

# A   Proof of the Sample Complexity Separation Theorem

As a warm-up, we prove Theorem 1 for $\mathbb{R}$ and $C = 2$ classes. We will use this as a building block for the general result.

## A.1   Warm-up: binary case

For this special case, the universe for the examples will be the real line $\mathbb{R}$, and we consider a binary classification task with a third category that only appears in the testing distribution. Let $\epsilon \in (0, 1/2)$ be a parameter. For the training distribution $\mu$, we define four regions:

1. **Positive, large probability.** Let $\mathcal{P}_0 = [1, 2]$, labeled as "+".

2. **Positive, small probability.** Let $\mathcal{P}_1 = [3, 4]$, labeled as "+".

3. **Negative, large probability.** Let $\mathcal{N}_0 = [-2, -1]$, labeled as "−".

4. **Negative, small probability.** Let $\mathcal{N}_1 = [-4, -3]$, labeled as "−".

To sample from the training distribution $\mu$, we first set $\ell \in \{-1, 1\}$ randomly with equal probability. Then, we choose $i \in \{0, 1\}$, where $i = 0$ with probability $1 - \epsilon$ and $i = 1$ with probability $\epsilon$. If $\ell = 1$, sample a point $x$ uniformly from $\mathcal{P}_i$, and otherwise, if $\ell = -1$, we sample uniformly from $\mathcal{N}_i$. Note that with probability $1 - \epsilon$, we have that $x \in \mathcal{P}_0 \cup \mathcal{N}_0$, while the probability of seeing any point in $\mathcal{P}_1 \cup \mathcal{N}_1$ is only $\epsilon$. Finally, let $\nu$ be the uniform distribution on $[-6, -5] \cup [5, 6]$, where for $x \sim \nu$, we label it as $\mathsf{sign}(x)$.

We first argue that nearest category generalization can be efficiently solved. During training time, if we see at least 32 samples from $\mu$, then with probability at least 99%, we will see samples from both $\mathcal{P}_0$ and $\mathcal{N}_0$, since $1 - \epsilon > 1/2$, and we see samples from each class with equal probability. Therefore, once we have at least one sample from each class, we can construct the classifier decides $\pm 1$ based on the midpoint of the training examples (which will be between $-2$ and $+2$ with good probability). Then, on the testing distribution $\nu$, we see that all points will be classified correctly with the label of their nearest neighbor in the support of $\mu$.

Turning to out-of-distribution detection, we claim that $\Omega(1/\epsilon)$ samples are necessary. Indeed, to distinguish whether a sample comes from $\nu$ or from $\mathcal{P}_1 \cup \mathcal{N}_1$, we must see at least one sample from each of $\mathcal{P}_1$ and $\mathcal{N}_1$, since the support of $\nu$ is unknown at training time. As the probability of sampling from $\mathcal{P}_1$ or $\mathcal{N}_1$ is only $\epsilon$, we will miss one of these regions with probability 99% if we have fewer than $t = 1/(100\epsilon)$ samples from $\mu$. Indeed, with probability $(1 - \epsilon)^t \geqslant e^{-\epsilon t} = e^{-.01} > 0.99$, we have that all the samples come from $\mathcal{P}_0 \cup \mathcal{N}_0$.

## A.2   General case

We now provide the proof of Theorem 1 for any number $C \geqslant 2$ of classes and for any $d \geqslant 1$ dimensional dataset in $\mathbb{R}^d$ with nearest neighbors measured in $\ell_2$ distance.

For $j \in \{1, 2, \ldots, C\}$ we define the following centers

$$a_0^j = 1 + 10j \qquad \text{and} \qquad a_1^j = 3 + 10j \qquad \text{and} \qquad a_2^j = 5 + 10j,$$

where we naturally embed them in $d$ dimensions by using these as the value of the first coordinate and setting the rest of the coordinates to be zero. In other words, we define $\mathbf{a}_i^j = a_i^j \cdot \mathbf{e}_1$, where $\mathbf{e}_1$ is the standard basis vector, so that $\mathbf{a}_i^j \in \mathbb{R}^d$.

Then, for $i \in \{0, 1, 2\}$ and $j \in \{1, 2, \ldots, C\}$, we define the following regions, which are cubes centered at the points defined above and have side length $1/\sqrt{d}$. Formally, we consider the $d$-dimensional cubes

$$\mathcal{A}_i^j = \left\{ \mathbf{a}_i^j + (x_1, x_2, \ldots, x_d) \mid 0 \leqslant x_k \leqslant 1/\sqrt{d} \right\}.$$

To sample from the training distribution $\mu$, we first choose $\ell \in \{1, 2, \ldots, C\}$ uniformly at random. Then, we choose $i \in \{0, 1\}$, where $i = 0$ with probability $1 - \epsilon$ and $i = 1$ with probability $\epsilon$. Given our choice of $\ell$, we

sample a point $\mathbf{x}$ uniformly from $\mathcal{A}_i^{\ell}$. Note that with probability $1 - \epsilon$, we have that $\mathbf{x} \in \bigcup_{j=1}^{C} \mathcal{A}_0^j$, while the probability of seeing any point in $\bigcup_{j=1}^{C} \mathcal{A}_1^j$ is only $\epsilon$. Finally, let $\nu$ be the uniform distribution on $\bigcup_{j=1}^{C} \mathcal{A}_2^j$. For both distributions, we label $\mathbf{x}$ as $j$ if it comes from $\mathcal{A}_i^j$ for any $i \in \{0, 1, 2\}$.

Notice that this definition with $j = 0$ corresponds to the positively labeled regions ($[1,2], [3,4], [5,6]$) from the proof of the binary case in the previous subsection. The probabilities are also the same when $C = 2$.

We explain the key properties of these regions, and then we prove the sample complexity results claimed in the theorem statement. First, for any $i \in \{0, 1, 2\}$ and $j \in \{1, 2, \ldots, C\}$, if $\mathbf{x}, \mathbf{y} \in \mathcal{A}_i^j$, then $\|\mathbf{x} - \mathbf{y}\|_2 \leqslant 1$ because each $\mathcal{A}_i^j$ is a cube with side length $1/\sqrt{d}$ in $\mathbb{R}^d$.

Next, consider $\mathbf{x} \in \mathcal{A}_2^j$. We claim that $\mathbf{x}$ is closer to $\mathcal{A}_0^j$ than to any point $\mathbf{z} \in \mathcal{A}_0^{j'} \cup \mathcal{A}_1^{j'}$ for any $j' \neq j$. To see this, we can check that the triangle inequality implies that

$$\min_{\mathbf{y} \in \mathcal{A}_0^j} \|\mathbf{x} - \mathbf{y}\|_2 \leqslant 4 + 1 = 5,$$

while, since the centers satisfy $|a_2^j - a_1^{j'}| > |a_2^j - a_0^{j'}| \geqslant 6$, we also have that for $j' \neq j$,

$$\min_{\mathbf{z} \in \mathcal{A}_0^{j'} \cup \mathcal{A}_1^{j'}} \|\mathbf{x} - \mathbf{z}\|_2 \geqslant 6.$$

As a consequence, we have that the nearest neighbor in $\ell_2$ distance for any point $\mathbf{x} \in \mathcal{A}_2^j$ has the same label $j$ as $\mathbf{x}$ does. In particular, this implies that we can solve the nearest category generalization problem for points sampled from $\nu$. To do so, we first sample $\Theta(C \log C)$ points from $\mu$, so that by a coupon collector argument, we see at least one point from $\mathcal{A}_0^j$ for each $j \in \{1, 2, \ldots, C\}$. Then, recall that $\nu$ is supported on the union of $\mathcal{A}_2^{j'}$ over $j' \in \{1, 2, \ldots, C\}$. By the above calculations, we have that the nearest neighbor for a point $\mathbf{x} \in \mathcal{A}_2^j$ is some point from either $\mathcal{A}_0^j$ or $\mathcal{A}_1^j$. Therefore, since we have sampled at least one point from $\mathcal{A}_0^j$, we can correctly determine that $\mathbf{x}$ has label $j$ by computing the nearest neighbor in our sampled points. To be more precise, we can compute the multi-class large-margin classifier, where we have sequential decision regions (corresponding roughly to the centers defined above), setting the decision boundaries to be equally spaced between samples from adjacent regions (i.e., the natural generalization of the 1D large-margin solution). Importantly, this solution does not require any extra knowledge of the support of $\mu$ and $\nu$ because it can be computed directly from the samples (and we have argued that with $\Theta(C \log C)$ samples, we will see all $C$ classes at least once).

We turn our attention to our lower bound, which is that we need at least $\Omega(C/\epsilon)$ samples to solve the OOD detection problem. More precisely, we provide a lower bound for the number of samples to guarantee that we see that least one point from each region $\mathcal{A}_1^j$ for each $j \in \{1, 2, \ldots, C\}$. This is a prerequisite for solving the OOD detection problem, because otherwise, we cannot tell whether a point comes from $\mu$ or $\nu$ without prior knowledge of the regions. For the lower bound, we use the same argument as in the binary case in the previous subsection. This implies that we need $\Omega(C/\epsilon)$ samples to see one from $\mathcal{A}_1^j$ for each fixed $j$ since the probability of sampling from this region is $\epsilon/C$ by the definition of $\mu$.

### A.3   Alternative generalizations

We could also use a "noisy one-hot encoding" to prove the theorem, replicating and rotating the 1D dataset $\log_2 C$ times, to get a subset of $\mathbb{R}^{\log_2 C}$ for $C$ classes. One dimension is non-zero for each point, and each dimension has points from two possible labels ($C$ total labels). Use $6C$ regions to define the low probability, high probability, and OOD regions (6 in each dimension with 3 for each class). Again, by a coupon collector argument, we will see some point from each of the high probability regions after $O(C \log C)$ samples. This enables nearest category generalization. On the other hand, for OOD detection, we need $\Omega(C/\epsilon)$ samples, where $\epsilon$ is the sample probability from a low probability region.

Instead of boxes, we could use Gaussian distributions with covariance $\sigma^2 I_d$ and means shifted by increments of a vector, spacing out the means by distance $\Omega(\sigma \sqrt{\log(d/\epsilon)})$ to get analogous guarantees. Similar ideas

work for Hamming distance on $\{0,1\}^d$; embed regions as intervals in the partial order along a path from $0^d$ to $1^d$, spacing them out to ensure 1-NN properties. In general, there are many metric spaces where we can provide a separation between nearest category generalization and OOD detection by correctly setting up the regions and sampling probabilities. Therefore, we believe it a general phenomena that nearest category generalization is a more tractable goal, in terms of sample complexity, than OOD detection.

# B More details for algorithm implementation

**Minimizing Equation (1).** Directly minimizing this loss function is challenging because of the second term; therefore, we make some approximations. We compute the inner maximization using the projected gradient descent algorithm (PGD) (Kurakin et al., 2016). PGD is initialized as: $\mathbf{x}_i^{(1)} = \mathbf{x}_i$, where $\mathbf{x}_i$ is the $i$-th training example. For iteration $t$, we take a gradient step on $\mathbf{x}_i'$ with step size $\alpha$ towards maximizing the KL divergence term (formally: $\mathbf{x}_i^{(t)} = \mathbf{x}_i^{(t-1)} + \alpha \nabla_{\mathbf{x}_i'} D_{\mathrm{KL}}(f_\theta(\mathbf{x}_i'), f_\theta(\mathbf{x}_i)))$, and then project $\mathbf{x}_i^{(t)}$ onto the region $P_i$. After $T$ iterations, we use $\mathbf{x}_i^{(T)}$ as the solution to the inner maximization and update the parameter $\theta$ by a stochastic gradient step on $\ell(f_\theta(\mathbf{x}_i), y_i) + D_{\mathrm{KL}}(f_\theta(\mathbf{x}_i^{(T)}), f_\theta(\mathbf{x}_i))$.

**Projecting onto different regions.** The projection needs to be computationally efficient as we need to project many points onto many regions for each update of the network. Projecting from a point to a ball is efficient as we can divide the norm of the point and multiply the radius of the ball. The projection onto an ellipsoid can be reduced to a second-order cone program, and the projection onto the sub-sampled sub-Voronoi region is a quadratic program. Sophisticated solvers for these two types of programs exist, but it can still be difficult when we have a large dataset or the data resides in a high-dimensional space. Fortunately, we only need an approximated solution for these projections. We adopt a binary search method to solve these programs approximately. We perform a binary search between the point that we want to project ($\mathbf{x}$) and the $i$-th training example ($\mathbf{x}_i$ that is in $P_i$). We use the point among all the points between $\mathbf{x}$ and $\mathbf{x}_i$ that is the closest to $\mathbf{x}$ but within the region $P_i$ as the projected point.

**Step size of PGD.** For both TRADES (uniform balls) and non-uniform balls, we set the step size of PGD to the robust radius $r$ and $\epsilon_i^{max}$ divided by 5. Different from balls, where the distance from the starting point ($\mathbf{x}_i$) to the boundary of the region is exactly the same for all directions, deciding the step size for the ellipsoid and sub-sampled sub-Voronoi region can be difficult. We heuristically set the step size for the ellipsoid to be the longest axis of each ellipsoid is divided by 5 As for sub-sampled sub-Voronoi regions, we set the step size as the distance from $\mathbf{x}_i$ to the furthest linear constraint divided by 5.

## B.1 Implementation details for each region

**Sub-sampled sub-Voronoi region.** We set the number of samples to 100 for MNIST and 50 for CIFAR10 and CIFAR100. Let $\mathbf{W}_j \mathbf{x} \leqslant \mathbf{h}_j$ be a linear constraint between example $\mathbf{x}_i$ and an example $\mathbf{x}_j$ where $y_i \neq y_j$, we add the shrinkage parameter $\lambda$ to the constraint as $\mathbf{W}_j \mathbf{x} \leqslant \mathbf{h}_j * \lambda * y_j$.

**Non-uniform radius ball.** The radius of the ball $\lambda \epsilon_i^{max}$ can be large when examples are far apart. Cheng et al. (2020); Sitawarin et al. (2020) report that enforcing smoothness in a large radius ball can be difficult and proposed methods that can diffuse some of these challenges. We adopt two methods from these works. First, for each example $\mathbf{x}_i$, we set its radius $\epsilon_i = 0$ and then gradually increase $\epsilon_i$ with a step size $\eta$ after each epoch. Second, if the prediction within the ball centered at $\mathbf{x}_i$ with radius $\epsilon_i$ is not smooth enough, we decrease $\epsilon_i$ by $\eta$. We set a threshold *thresh* to determine whether it is smooth enough. The pseudocode is in Algorithm 1.

**Ellipsoid.** In practice, we set the number of differently-labeled samples $k = 100$. Let $s_i, \mathbf{V}_i$ be the top $\frac{k}{2}$ singular values and principal components respectively, We search for the shrinkage parameter $\lambda_i$ between 1 and 500 with binary search and find the largest $\lambda_i$ such that the ellipsoid include at most 5% of the $k$ sampled examples. Then, we divide $\lambda_i$ by 2 to make sure the ellipsoid from another point does not overlap with the current ellipsoid.

---

**Algorithm 1** $\{\mathbf{x}_i, y_i\}_{i=1}^N, \lambda, \beta, \eta, thresh, T$

---

$\epsilon_i^{max} = \lambda \cdot \min_{\mathbf{x}_j \in \mathcal{X}^{\neq y_i}} \frac{1}{2}\mathsf{dist}(\mathbf{x}_i, \mathbf{x}_j)$
$\epsilon_i \leftarrow 0. \quad \forall i \in [N]$
**for** # *epoch* **do**
 **for** $i = 1..N$ **do**
  $\epsilon_i \leftarrow \epsilon_i + \eta$
  $\epsilon_i = max(\epsilon_i, \epsilon_i^{max})$
  $\delta_i \leftarrow 0$
  **for** $j = 1..T$ **do**
   $\delta_i \leftarrow \alpha \cdot sign(\nabla_{\delta_i} D_{\mathrm{KL}}(f_\theta(\mathbf{x}_i + \delta_i), f_\theta(\mathbf{x}_i))$
   project $\delta_i$ onto $\mathcal{B}(0, \epsilon_i)$
  **end for**
  **if** $D_{\mathrm{KL}}(f_\theta(\mathbf{x}_i + \delta_i), f_\theta(\mathbf{x}_i)) > thresh$ **then**
   $\epsilon_i \leftarrow \epsilon_i - 2 * \eta$
  **end if**
  update $\theta$ to minimize $\ell(f_\theta(\mathbf{x}_i), y_i) + \beta * D_{\mathrm{KL}}(f_\theta(\mathbf{x}_i + \delta_i), f_\theta(\mathbf{x}_i))$
 **end for**
**end for**

---

## C  Detailed experiment setups

The experiments are performed on 6 NVIDIA GeForce RTX 2080 Ti and 2 RTX 3080 GPUs located on three servers. Two of the servers have Intel Core i9 9940X and 128GB of RAM and the other one has AMD Threadripper 3960X and 256GB of RAM. We compute nearest neighbors using FAISS[2] (Johnson et al., 2017), and all neural networks are implemented under the PyTorch framework[3] (Paszke et al., 2019). The code for all the experiments can be found at `https://github.com/yangarbiter/nearest-category-generalization`.

**Algorithm implementations.** For C&W attack algorithm (Carlini & Wagner, 2017), we use the implementation by foolbox[4] (Rauber et al., 2017). For TRADES (Zhang et al., 2019), we also use the implementation From the original author[5].

**Datasets.** All datasets used in our paper can be found in publicly available urls. MNIST can be found in `http://yann.lecun.com/exdb/mnist/`, CIFAR10 and CIFAR100 can be found in `https://www.cs.toronto.edu/~kriz/cifar.html`, ImgNet can be found in `https://www.image-net.org/`.

**Architechtures.** We consider the convolutional neural network (CNN)[6], wider residual network (WRN-40-10) (Zagoruyko & Komodakis, 2016), ResNet50 (He et al., 2016) for our experiments in the pixel space.

**Optimizers.** We consider stochastic gradient descent (SGD) and Adam (Kingma & Ba, 2014) as the optimizers.

**MNIST setup.** We use the CNN used by Zhang et al. (2019) for training neural networks in the pixel space. The learning rate is decreased by a factor of 0.1 on the 40-th, 50-th, and 60-th epoch. We use the output of the last convolutional CNN output as the extracted feature.

**CIFAR10, CIFAR100, ImgNet100 setup.** For CIFAR10 and CIFAR100, we use Wider ResNet (WRN-40-10) (Zagoruyko & Komodakis, 2016) for training neural networks in the pixel space. For ImgNet100, we use ResNet50 (He et al., 2016) for training neural networks in the pixel space. The learning rate is decreased by a factor of 0.1 on the 40-th, 50-th, and 60-th epoch. For ImgNet100, we normalize the data by subtracting the mean (0.485, 0.456, 0.406) and standard deviation (0.229, 0.224, 0.225).

---

[2]code and license can be found in `https://github.com/facebookresearch/faiss`

[3]code and license can be found in `https://github.com/pytorch/pytorch`

[4]code and license can be found in `https://github.com/bethgelab/foolbox`

[5]code and license can be found in `https://github.com/yaodongyu/TRADES`

[6]CNN is retrieved from the public repository of TRADES (Zhang et al., 2019) `https://github.com/yaodongyu/TRADES/blob/master/models/small_cnn.py`

| dataset | MNIST | CIFAR10 | CIFAR100 | ImgNet100 |
|---|---|---|---|---|
| network structure | CNN | WRN-40-10 | WRN-40-10 | ResNet50 |
| optimizer | SGD | Adam | Adam | Adam |
| batch size | 128 | 64 | 64 | 128 |
| momentum | 0.9 | - | - | - |
| epochs | 70 | 70 | 70 | 70 |
| initial learning rate | 0.01 | 0.01 | 0.01 | 0.01 |
| # train examples | 60000 | 50000 | 50000 | 126689 |
| # test examples | 10000 | 10000 | 10000 | 5000 |
| # classes | 10 | 10 | 20 | 100 |

Table 5: Experimental setup for training in the pixel space. No weight decay is applied.

**Adversarial attack algorithms.** For the adversarial attack algorithms used to find the closest adversarial examples, we use a mixture of projected gradient descent (PGD) (Madry et al., 2017), Brendel Bethge attack (Brendel et al., 2019), boundary attack (Brendel et al., 2017), multi-targeted attack (Kwon et al., 2018), Sign-Opt (Cheng et al., 2019) and C&W algorithm (Carlini & Wagner, 2017).

### C.0.1 Setups for experiments in the feature space

**Architechtures.** We use small networks to compute the embedding into the feature space (training without the unseen class for the unseen class experiments). We continue to use a CNN or WRN for training and prediction. For the feature space, for MNIST, CIFAR10, and CIFAR100, we train a multi-layer-perceptron (MLP) with two hidden layers each with 256 neurons and ReLU as the activation function in the feature space. For ImgNet100, we train an MLP with two hidden layers each with 1024 neurons and ReLU as the activation function in the feature space. For all four datasets, we use SGD as the optimizer with an initial learning rate of 0.01 and a momentum of 0.9.

## D  Additional experiment results

### D.1  Synthetic data

Here, we present several more results on synthetic data to showcase how enforcing smoothness onto different regions can change the geometry of the decision boundary. From Figure 7, we see that natural training, in general, has a more irregular decision boundary while the other methods that enforce local smoothing have more vertically straight boundaries.

### D.2  Extension of Table 3

Table 6 shows the result of four more dataset on top of Table 3. Similar conclusions can be made on these datasets.

### D.3  Interaction between NCG score and accuracy on corrupted data

Table 7 shows an example of the interaction between NCG score and accuracy on the Gaussian noise corrupted data. From the table, we see three things: (1) NCG scores are above chance level, (2) test accuracies on NCG correct examples are higher than the test accuracies on NCG incorrect examples, and (3) the difference between natural training and TRADES are small in the feature space in comparison with the pixel space. This is a typical result, and similar results can be found in other corruption types. These observations are similar to those mentioned in the main text.

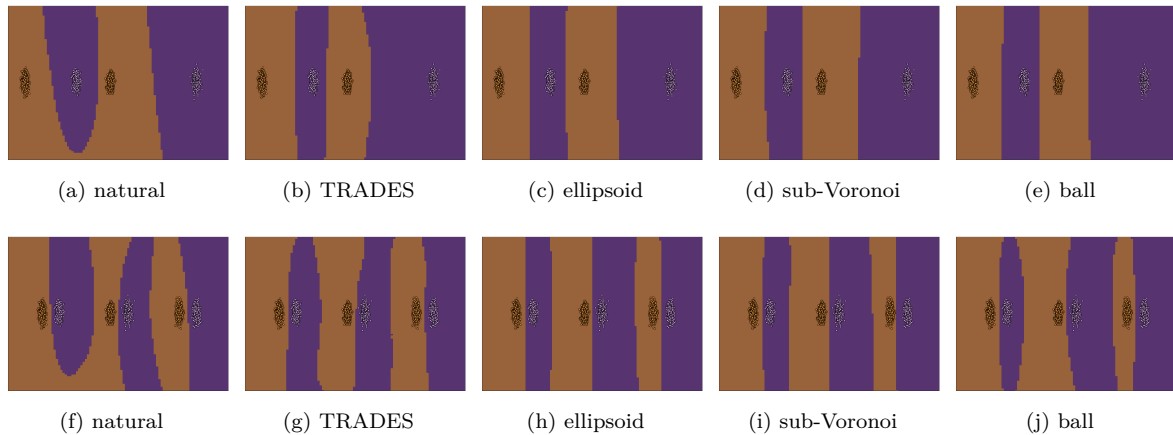

|     | (a) natural | (b) TRADES | (c) ellipsoid | (d) sub-Voronoi | (e) ball |
|     | (f) natural | (g) TRADES | (h) ellipsoid | (i) sub-Voronoi | (j) ball |

Figure 7: The geometry of the decision boundary with different algorithms.

Table 6: The training, testing, and NCG score of neural networks trained by enforcing smoothness on different regions. Here we consider MNIST with five and eight, CIFAR10 with *truck*, and CIFAR100 with *large man-made outdoor things* as the unseen categories.

|             | trn acc. | tst acc. | NCG score | trn acc. | tst acc. | NCG score |
|-------------|----------|----------|-----------|----------|----------|-----------|
|             |          | MNIST-wo5 |          |          | MNIST-wo8 |          |
| sub-voronoi | 0.982    | 0.982    | 0.628     | 0.982    | 0.981    | 0.513     |
| ellipsoid   | 0.984    | 0.983    | **0.629** | 0.983    | 0.981    | **0.515** |
| ball        | 0.978    | 0.976    | 0.625     | 0.979    | 0.976    | 0.513     |
| TRADES      | 0.988    | 0.987    | 0.618     | 0.987    | 0.987    | 0.497     |
| nat         | 1.000    | 0.995    | 0.505     | 1.000    | 0.994    | 0.416     |
|             |          | CIFAR10-wo9 |        |          | CIFAR100-wo9 |        |
| sub-voronoi | 0.414    | 0.409    | **0.296** | 0.582    | 0.455    | **0.517** |
| ellipsoid   | 0.519    | 0.504    | 0.226     | 0.658    | 0.508    | 0.449     |
| ball        | 0.813    | 0.639    | 0.244     | 0.896    | 0.472    | 0.473     |
| TRADES      | 0.778    | 0.641    | 0.245     | 0.867    | 0.515    | 0.475     |
| nat         | 1.000    | 0.885    | 0.145     | 1.000    | 0.703    | 0.235     |

### D.3.1   Control for distance

From Figure 6, we observe that a closer distance to the closest training example leads to a higher NCG score. However, the distance to the closest training example could be a confounding factor that leads to the observation of "test accuracies on NCG correct examples are higher than the test accuracies on NCG incorrect examples". To test whether this is the case, we gathered all examples with the same corruption type but different corruption levels, and we binned these examples into five equal-width bins based on their distance to the closest training example. We then measure, in each bin, how many corruption sets have their test accuracies on NCG correct examples significantly higher than the test accuracies on NCG incorrect examples.

The results are shown in Figure 8, and we labeled each bin from one to five based on their average distance to the closest training examples from the closest to farthest. From the result, we see that except ImgNet100 in the pixel space, in all other cases, the results are similar between different bins. This means that, in general, the distance to the closest training examples does not affect whether "test accuracies on NCG correct examples are higher than the test accuracies on NCG incorrect examples" or not.

Table 7: Here we show models trained on CIFAR10 and CIFAR100 and evaluate them on the gaussian noise corrupted data. The NCG score, test accuracy, the test accuracy on the NCG correct corrupted examples, the test accuracy on the NCG incorrect corrupted examples, and the distance to the closest training example.

| | model | | natural | | | | TRADES(2) | | | |
| | dataset | corruption level | tst acc. | NCG incorrect tst acc. | NCG correct tst acc. | NCG score | tst acc. | NCG incorrect tst acc. | NCG correct tst acc. | NCG score |
|---|---|---|---|---|---|---|---|---|---|---|
| pixel | C10 | 1 | 0.76 | 0.70 | 0.88 | 0.34 | 0.71 | 0.67 | 0.78 | 0.40 |
| | | 2 | 0.63 | 0.54 | 0.82 | 0.30 | 0.71 | 0.66 | 0.78 | 0.39 |
| | | 3 | 0.48 | 0.39 | 0.75 | 0.26 | 0.70 | 0.65 | 0.77 | 0.39 |
| | | 4 | 0.41 | 0.32 | 0.70 | 0.24 | 0.69 | 0.63 | 0.77 | 0.38 |
| | | 5 | 0.36 | 0.27 | 0.66 | 0.22 | 0.68 | 0.63 | 0.77 | 0.38 |
| | C100 | 1 | 0.63 | 0.56 | 0.84 | 0.25 | 0.52 | 0.43 | 0.72 | 0.30 |
| | | 2 | 0.55 | 0.47 | 0.79 | 0.24 | 0.51 | 0.43 | 0.71 | 0.30 |
| | | 3 | 0.47 | 0.39 | 0.74 | 0.23 | 0.51 | 0.42 | 0.71 | 0.30 |
| | | 4 | 0.44 | 0.36 | 0.71 | 0.22 | 0.50 | 0.42 | 0.71 | 0.30 |
| | | 5 | 0.40 | 0.33 | 0.67 | 0.21 | 0.50 | 0.41 | 0.71 | 0.29 |
| | I | 1 | 0.42 | 0.41 | 0.68 | 0.04 | 0.36 | 0.35 | 0.51 | 0.06 |
| | | 2 | 0.34 | 0.33 | 0.64 | 0.03 | 0.36 | 0.35 | 0.53 | 0.05 |
| | | 3 | 0.22 | 0.21 | 0.49 | 0.03 | 0.34 | 0.33 | 0.49 | 0.05 |
| | | 4 | 0.12 | 0.11 | 0.24 | 0.02 | 0.30 | 0.30 | 0.45 | 0.05 |
| | | 5 | 0.04 | 0.04 | 0.07 | 0.02 | 0.22 | 0.22 | 0.34 | 0.04 |
| feature | C10 | 1 | 0.74 | 0.39 | 0.78 | 0.89 | 0.72 | 0.32 | 0.77 | 0.89 |
| | | 2 | 0.59 | 0.35 | 0.64 | 0.85 | 0.56 | 0.23 | 0.62 | 0.85 |
| | | 3 | 0.45 | 0.33 | 0.48 | 0.82 | 0.40 | 0.19 | 0.45 | 0.83 |
| | | 4 | 0.39 | 0.33 | 0.40 | 0.81 | 0.35 | 0.20 | 0.38 | 0.83 |
| | | 5 | 0.34 | 0.28 | 0.35 | 0.82 | 0.31 | 0.18 | 0.33 | 0.83 |
| | C100 | 1 | 0.60 | 0.25 | 0.72 | 0.74 | 0.62 | 0.29 | 0.71 | 0.78 |
| | | 2 | 0.51 | 0.24 | 0.63 | 0.68 | 0.53 | 0.29 | 0.62 | 0.74 |
| | | 3 | 0.43 | 0.23 | 0.54 | 0.64 | 0.44 | 0.25 | 0.53 | 0.69 |
| | | 4 | 0.40 | 0.22 | 0.51 | 0.63 | 0.40 | 0.23 | 0.49 | 0.67 |
| | | 5 | 0.37 | 0.21 | 0.46 | 0.61 | 0.37 | 0.21 | 0.46 | 0.65 |
| | I | 1 | 0.22 | 0.18 | 0.44 | 0.15 | 0.21 | 0.18 | 0.41 | 0.16 |
| | | 2 | 0.19 | 0.16 | 0.36 | 0.14 | 0.18 | 0.15 | 0.34 | 0.15 |
| | | 3 | 0.14 | 0.12 | 0.26 | 0.14 | 0.13 | 0.11 | 0.21 | 0.17 |
| | | 4 | 0.09 | 0.08 | 0.16 | 0.13 | 0.08 | 0.07 | 0.14 | 0.16 |
| | | 5 | 0.05 | 0.04 | 0.08 | 0.14 | 0.04 | 0.03 | 0.08 | 0.14 |

## D.4 Ablation study

To showcases that our discovery are not only observed in our training setup but also extends in other scenarios, we repeat our experiment with a different network architecture in Appendix D.4.1 and models trained by other researchers in Appendix D.4.2.

### D.4.1 A different architecture

We repeat the experiment with a different network architecture – DenseNet161 (Huang et al., 2017). Table 9 shows the training, testing, and NCG scores. We see that even with a different architecture, robust models still have a higher NCG score than naturally trained models.

Table 8: Number of cases where the NCG correct examples have a **significantly** higher test accuracy than the NCG incorrect examples. 12/15 means that out of the 15 corrupted sets, 12 of them pass the t-test with 95% confidence level.

|  | corrupt lvl | feature | | | pixel | | |
|---|---|---|---|---|---|---|---|
|  |  | C10 | C100 | I | C10 | C100 | I |
| TRADES(2) | 1 | 18/18 | 18/18 | 12/15 | 18/18 | 18/18 | 9/15 |
|  | 2 | 18/18 | 18/18 | 14/15 | 18/18 | 18/18 | 11/15 |
|  | 3 | 18/18 | 18/18 | 14/15 | 17/18 | 18/18 | 10/15 |
|  | 4 | 18/18 | 18/18 | 14/15 | 14/18 | 18/18 | 10/15 |
|  | 5 | 18/18 | 18/18 | 15/15 | 14/18 | 18/18 | 3/15 |
| natural | 1 | 18/18 | 18/18 | 13/15 | 18/18 | 18/18 | 12/15 |
|  | 2 | 18/18 | 18/18 | 13/15 | 18/18 | 18/18 | 12/15 |
|  | 3 | 18/18 | 18/18 | 13/15 | 18/18 | 18/18 | 12/15 |
|  | 4 | 18/18 | 18/18 | 14/15 | 17/18 | 18/18 | 12/15 |
|  | 5 | 18/18 | 18/18 | 15/15 | 18/18 | 18/18 | 9/15 |

Table 9: Results with DenseNet161 on CIFAR10 and CIFAR100.

|  |  | natural | AT(2) | TRADES(2) |
|---|---|---|---|---|
| CIFAR10-wo0 | train acc. | 1.000 | 0.781 | 0.876 |
|  | test acc. | 0.839 | 0.637 | 0.640 |
|  | NCG score | 0.342 | 0.487 | 0.521 |
| CIFAR100-wo0 | train acc. | 1.000 | 0.886 | 0.557 |
|  | test acc. | 0.608 | 0.500 | 0.441 |
|  | NCG score | 0.173 | 0.225 | 0.271 |

### D.4.2 Pretrained models on corrupted data

To verify that our observations on corrupted data can also be observed by models trained by others, we download pretrained models from `https://github.com/MadryLab/robustness/tree/master/robustness` by Engstrom et al. (2019). We cannot repeat the experiment for unseen classes since these models are trained on the original CIFAR10.

For models in the features space, we follow the same setup as in the feature space of CIFAR10, which trains a multi-layer perceptron on the CNN feature space, but in the feature space of the pretrained model. Table 10 shows three things: (1) the number of cases (out of 90 corrupted sets) where robust models that have an NCG score higher than naturally trained models, (2) the average difference in NCG score between robust networks and naturally trained networks (over the 90 corrupted sets), and (3) the average ratio in NCG score between robust networks and naturally trained networks. From the table, we see that robust models with a large enough robust radius in general have a larger NCG score than naturally trained models.

### D.5 NCG score on in-distribution data

In Section 3, we claim that the robust networks are more likely to classify OOD data with the class label of the nearest training input. One question is, does this phenomena also happen on in-distribution data? To answer this question, we measures the NCG score on in-distribution data (which is the test accuracy of a 1-nearest neighbor classifier).

The results are in Table 11. From the table, we see that robust training usually produces a model that has a slightly higher NCG score. However, it seems that these increases on in-distribution NCG score is small. A natural question is, compare to OOD data, these increases on in-distribution NCG score are larger or smaller.

Table 10: In both pixel and feature space, among the 90 corrupted sets for CIFAR10, the first columns shows the number of robust models that have an NCG score higher than naturally trained network. The second and third column shows the average difference and ratio of the NCG score of the robust models and naturally trained networks (average over the NCG scores on the 90 corrupted sets).

|  |  | robust > natural counts | difference | ratio |
|---|---|---|---|---|
|  |  | pixel | | |
| CIFAR10 | AT(0.25) | 51/90 | $0.00 \pm 0.05$ | $1.14 \pm 0.04$ |
|  | AT(0.5) | 86/90 | $0.14 \pm 0.10$ | $3.27 \pm 0.55$ |
|  | AT(1.0) | 88/90 | $0.18 \pm 0.06$ | $3.09 \pm 0.22$ |
|  |  | feature | | |
| CIFAR10 | AT(1.0) | 70/90 | $0.00 \pm 0.00$ | $1.01 \pm 0.00$ |
|  | TRADES(2) | 55/90 | $0.00 \pm 0.00$ | $1.00 \pm 0.00$ |
|  | TRADES(4) | 52/90 | $0.00 \pm 0.00$ | $1.00 \pm 0.00$ |
|  | TRADES(8) | 55/90 | $0.00 \pm 0.00$ | $1.00 \pm 0.00$ |

Table 11: The in-distribution NCG score on different dataset and training methods.

|  |  | natural | TRADES(2) | TRADES(4) | TRADES(8) | AT(2) |
|---|---|---|---|---|---|---|
|  | M-0 | 0.969 | 0.971 | 0.947 | 0.969 | 0.970 |
|  | M-1 | 0.969 | 0.970 | 0.967 | 0.971 | 0.971 |
|  | M-2 | 0.971 | 0.972 | 0.954 | 0.972 | 0.973 |
|  | M-3 | 0.975 | 0.976 | 0.960 | 0.976 | 0.976 |
|  | M-4 | 0.973 | 0.976 | 0.963 | 0.974 | 0.975 |
|  | M-5 | 0.973 | 0.975 | 0.963 | 0.975 | 0.974 |
|  | M-6 | 0.968 | 0.971 | 0.951 | 0.969 | 0.970 |
|  | M-7 | 0.973 | 0.974 | 0.955 | 0.974 | 0.974 |
|  | M-8 | 0.974 | 0.974 | 0.962 | 0.975 | 0.975 |
| in-dist NCG | M-9 | 0.975 | 0.977 | 0.960 | 0.976 | 0.977 |
|  | C10-0 | 0.357 | 0.410 | 0.418 | 0.381 | 0.403 |
|  | C10-4 | 0.379 | 0.412 | 0.408 | 0.352 | 0.423 |
|  | C10-9 | 0.371 | 0.405 | 0.407 | 0.378 | 0.418 |
|  | C100-0 | 0.271 | 0.316 | 0.302 | 0.292 | 0.311 |
|  | C100-4 | 0.261 | 0.300 | 0.295 | 0.279 | 0.295 |
|  | C100-9 | 0.267 | 0.313 | 0.302 | 0.291 | 0.307 |
|  | I-0 | 0.038 | 0.059 | 0.058 | 0.055 | 0.059 |
|  | I-1 | 0.043 | 0.058 | 0.060 | 0.075 | 0.057 |
|  | I-2 | 0.048 | 0.057 | 0.058 | 0.066 | 0.057 |

In Table 12, we make such comparison. We count the number of cases (same dataset with different held-out classes) where the increase in NCG score for in-distribution data is large than the increase on OOD data by switching from natural training to robust training. We then categorize these results by the dataset and robust training method. From the result, we see that for most of the time, the increase in NCG score on OOD data is larger than on in-distribution data. This suggest that the phenomenon of NCG is more prominent on OOD than on in-distribution data.

## D.6 Additional results

For completeness, we show additional results to the tables or figures in the main paper.

Table 12: For both in- and out-of-distribution data, the NCG score increase as we switch the natural training to robust training. However, the magnitude of the increase is larger for OOD data. This table shows the The number of cases where the robust models with higher in-distribution NCG score than natural training. For MNIST we check 10 unseen classes, and for CIFAR10, CIFAR100, and ImgNet100, we use 3 unseen classes. 10/10 means that out of the 10 unseen classes, all 10 models have higher NCG score.

|            | M     | C10  | C100 | I    |
|------------|-------|------|------|------|
| TRADES(2)  | 0/10  | 0/3  | 0/3  | 3/3  |
| TRADES(4)  | 2/10  | 0/3  | 0/3  | 1/3  |
| TRADES(8)  | 2/10  | 0/3  | 0/3  | 1/3  |
| AT(2)      | 0/10  | 0/3  | 0/3  | 3/3  |

### D.6.1 Distance to the closest training examples of corrupted data

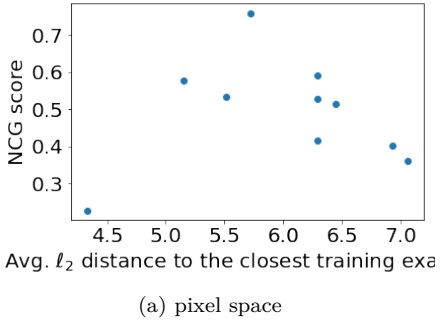

(a) pixel space

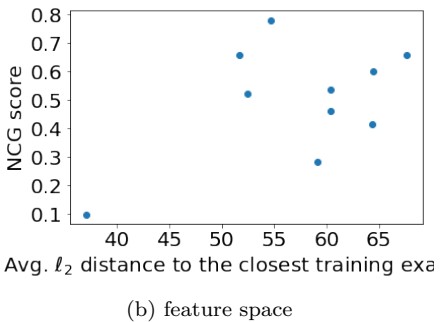

(b) feature space

Figure 8: NCG score of a naturally trained model as a function of the $\ell_2$ distance to the closest training example for MNIST with different unseen classes.

Tables 13 to 15 show the average distance to the closest training examples for each corruption type and level.

### D.6.2 NCG scores

In Figure 8, for each unseen class in MNIST with naturally trained models, we show the NCG score and the average distance of the examples in the unseen class to the closest training example. Tables 17 to 20 extends Table 1 with the full table of different dataset unseen class pairs in MNIST, CIFAR10, CIFAR100, and ImgNet100.

### D.6.3 NCG score vs. the distance to the closest training example

Figures 9 and 10 shows the NCG score and the distance to the closest training example for MNIST, CIFAR10, and CIFAR100 in both pixel and feature space. We can see that, in general, the NCG score is higher when in- and out-of-distribution examples are closer to each other.

### D.7 Additional results for corrupted data

**Robust models on corrupted data.** In the pixel space, on average (over the 90 and 75 corrupted sets), robust models have a NCG score that is $1.35 \pm .02$, $1.36 \pm .03$, and $1.66 \pm .04$ times higher than naturally trained models for CIFAR10, CIFAR100, and ImgNet100 respectively. In the feature space, we still find that **all** the 255 corruption sets have an NCG score above chance level, but the NCG scores of the robust models are closer to the naturally trained models. For CIFAR100, we still observe that **all** robust models have an NCG score higher than the naturally trained models. But for CIFAR10, we find that on only 42 (out of 90) corrupted sets, TRADES(2) models have a higher NCG score than naturally trained models. The

Table 13: CIFAR10 The average $\ell_2$ distance in the pixel space to the closest training example. A corruption level 0 means no corruption is applied, which represents the original test data.

| | corruption level | | | | | |
|---|---|---|---|---|---|---|
| | 0 | 1 | 2 | 3 | 4 | 5 |
| gaussian | 9.21 | 9.46 | 9.75 | 10.13 | 10.35 | 10.58 |
| impulse | 9.21 | 9.67 | 10.10 | 10.50 | 11.27 | 11.97 |
| shot | 9.21 | 9.35 | 9.49 | 9.89 | 10.10 | 10.49 |
| defocus | 9.21 | 9.01 | 8.73 | 8.51 | 8.38 | 8.08 |
| motion | 9.21 | 8.73 | 8.48 | 8.27 | 8.27 | 8.08 |
| zoom | 9.21 | 8.61 | 8.54 | 8.45 | 8.38 | 8.27 |
| glass | 9.21 | 9.29 | 9.21 | 8.85 | 9.28 | 8.87 |
| snow | 9.21 | 9.48 | 9.76 | 10.02 | 10.15 | 10.03 |
| fog | 9.21 | 8.02 | 7.06 | 6.63 | 6.50 | 6.50 |
| contrast | 9.21 | 7.45 | 5.68 | 4.96 | 4.28 | 3.44 |
| pixelate | 9.21 | 9.14 | 9.07 | 9.04 | 8.94 | 8.81 |
| brightness | 9.21 | 9.34 | 9.49 | 9.62 | 9.70 | 9.69 |
| elastic_transform | 9.21 | 8.93 | 8.84 | 8.65 | 8.60 | 8.59 |
| gaussian_blur | 9.21 | 9.02 | 8.51 | 8.33 | 8.17 | 7.88 |
| jpeg_compression | 9.21 | 9.20 | 9.17 | 9.16 | 9.14 | 9.13 |
| saturate | 9.21 | 9.03 | 9.11 | 9.68 | 10.18 | 10.59 |
| spatter | 9.21 | 9.36 | 9.59 | 9.92 | 9.73 | 10.02 |
| speckle_noise | 9.21 | 9.35 | 9.59 | 9.75 | 10.12 | 10.56 |

Table 14: CIFAR100 $\ell_2$ distance in the pixel space. A corruption level 0 means no corruption is applied, which represents the original test dat

| | 0 | 1 | 2 | 3 | 4 | 5 |
|---|---|---|---|---|---|---|
| gaussian | 9.18 | 9.42 | 9.71 | 10.07 | 10.29 | 10.51 |
| impulse | 9.18 | 9.66 | 10.11 | 10.53 | 11.32 | 12.04 |
| shot | 9.18 | 9.32 | 9.46 | 9.84 | 10.04 | 10.42 |
| defocus | 9.18 | 8.99 | 8.72 | 8.52 | 8.40 | 8.12 |
| motion | 9.18 | 8.72 | 8.49 | 8.30 | 8.29 | 8.12 |
| zoom | 9.18 | 8.62 | 8.53 | 8.44 | 8.37 | 8.24 |
| glass | 9.18 | 9.28 | 9.21 | 8.86 | 9.30 | 8.90 |
| snow | 9.18 | 9.40 | 9.62 | 9.86 | 9.93 | 9.72 |
| fog | 9.18 | 8.03 | 7.07 | 6.62 | 6.47 | 6.43 |
| contrast | 9.18 | 7.47 | 5.75 | 5.02 | 4.32 | 3.41 |
| pixelate | 9.18 | 9.12 | 9.06 | 9.03 | 8.93 | 8.81 |
| brightness | 9.18 | 9.30 | 9.42 | 9.51 | 9.56 | 9.47 |
| elastic_transform | 9.18 | 8.99 | 8.89 | 8.70 | 8.64 | 8.61 |
| gaussian_blur | 9.18 | 9.00 | 8.52 | 8.34 | 8.19 | 7.92 |
| jpeg_compression | 9.18 | 9.17 | 9.13 | 9.12 | 9.10 | 9.08 |
| saturate | 9.18 | 8.88 | 8.95 | 9.69 | 10.13 | 10.47 |
| spatter | 9.18 | 9.33 | 9.55 | 9.86 | 9.78 | 10.09 |
| speckle_noise | 9.18 | 9.32 | 9.56 | 9.71 | 10.08 | 10.52 |

average improvement over the naturally trained models in NCG score goes down to $1.00 \pm .00$, $1.07 \pm .00$, and $1.09 \pm .01$ times for CIFAR10, CIFAR100, and ImgNet100 respectively.

Table 15: ImgNet100 $\ell_2$ distance in the pixel space. A corruption level 0 means no corruption is applied, which represents the original test dat

|  | corruption level | | | | | |
|---|---|---|---|---|---|---|
|  | 0 | 1 | 2 | 3 | 4 | 5 |
| contrast | 58.92 | 81.52 | 68.49 | 56.77 | 47.79 | 45.10 |
| glass | 58.92 | 157.96 | 154.18 | 149.79 | 145.57 | 138.42 |
| shot | 58.92 | 164.22 | 164.39 | 163.95 | 162.02 | 160.25 |
| jpeg | 58.92 | 163.96 | 163.95 | 163.85 | 163.82 | 163.98 |
| impulse | 58.92 | 162.38 | 160.93 | 159.38 | 154.99 | 149.85 |
| elastic | 58.92 | 162.32 | 162.45 | 162.30 | 162.10 | 161.49 |
| zoom | 58.92 | 153.57 | 150.31 | 146.04 | 143.64 | 139.73 |
| frost | 58.92 | 153.25 | 131.21 | 120.42 | 117.39 | 112.21 |
| defocus | 58.92 | 154.88 | 151.06 | 144.13 | 139.73 | 133.87 |
| brightness | 58.92 | 167.31 | 167.13 | 163.70 | 157.28 | 149.42 |
| snow | 58.92 | 169.97 | 163.99 | 169.25 | 166.42 | 145.04 |
| pixelate | 58.92 | 162.17 | 161.71 | 159.82 | 157.66 | 156.16 |
| motion | 58.92 | 157.85 | 152.94 | 146.59 | 140.29 | 135.60 |
| gaussian | 58.92 | 163.65 | 163.30 | 161.90 | 158.52 | 152.48 |
| fog | 58.92 | 95.21 | 91.82 | 95.09 | 100.12 | 103.61 |

Table 16: The test accuracy of a 1-nearest neighbor classifier in the feature space 12 different datasets.

| M-0 | M-4 | M-9 | C10-0 | C10-4 | C10-9 | C100-0 | C100-4 | C100-9 | I-0 | I-1 | I-2 |
|---|---|---|---|---|---|---|---|---|---|---|---|
| 0.99 | 0.99 | 0.99 | 0.89 | 0.88 | 0.88 | 0.73 | 0.73 | 0.71 | 0.14 | 0.14 | 0.14 |

### D.7.1 Additional results on the slope of corrupted test accuracy

**NCG score.** Repeating the same experiment with NCG score, we find similar results as well. In the pixel space, for CIFAR10 and CIFAR100, the slope of naturally trained models are significantly smaller than TRADES(2) on 15 and 14 (out of 18) corruption types. For ImgNet100, 6 out of 15 corruption types pass the test. The other 9 corruption types are not significant (they did not accept or reject the hypothesis). In the feature space, we also test whether the slopes of robust and naturally trained models are different. For CIFAR10 and CIFAR100, 17 and 15 (out of 18) corruption types, respectively, are not significantly different. For ImgNet100, 13 out of 15 corruption types are not significant.

For reference, we show four selected corruption types on each dataset in Figure 11.

Table 17: The train, test, and NCG scores of 10 MNIST datasets and 5 training methods in the pixel space.

|  |  | natural | AT(2) | TRADES(2) | TRADES(4) | TRADES(8) |
|---|---|---|---|---|---|---|
| MNIST-wo0 | train acc. | 1.000 | 0.993 | 0.987 | 0.954 | 0.997 |
|  | test acc. | 0.995 | 0.990 | 0.985 | 0.956 | 0.995 |
|  | NCG score | 0.390 | 0.457 | 0.457 | 0.485 | 0.402 |
| MNIST-wo1 | train acc. | 1.000 | 0.994 | 0.987 | 0.975 | 0.997 |
|  | test acc. | 0.995 | 0.991 | 0.987 | 0.974 | 0.994 |
|  | NCG score | 0.273 | 0.451 | 0.355 | 0.528 | 0.259 |
| MNIST-wo2 | train acc. | 1.000 | 0.993 | 0.988 | 0.958 | 0.997 |
|  | test acc. | 0.994 | 0.990 | 0.987 | 0.962 | 0.994 |
|  | NCG score | 0.402 | 0.532 | 0.529 | 0.520 | 0.452 |
| MNIST-wo3 | train acc. | 1.000 | 0.994 | 0.989 | 0.962 | 0.997 |
|  | test acc. | 0.995 | 0.992 | 0.988 | 0.964 | 0.994 |
|  | NCG score | 0.564 | 0.659 | 0.667 | 0.592 | 0.538 |
| MNIST-wo4 | train acc. | 1.000 | 0.994 | 0.988 | 0.963 | 0.997 |
|  | test acc. | 0.995 | 0.991 | 0.987 | 0.966 | 0.995 |
|  | NCG score | 0.760 | 0.766 | 0.810 | 0.758 | 0.749 |
| MNIST-wo5 | train acc. | 1.000 | 0.993 | 0.988 | 0.965 | 0.997 |
|  | test acc. | 0.995 | 0.990 | 0.987 | 0.965 | 0.995 |
|  | NCG score | 0.505 | 0.611 | 0.618 | 0.616 | 0.537 |
| MNIST-wo6 | train acc. | 1.000 | 0.993 | 0.987 | 0.959 | 0.997 |
|  | test acc. | 0.995 | 0.991 | 0.987 | 0.962 | 0.995 |
|  | NCG score | 0.515 | 0.551 | 0.556 | 0.505 | 0.538 |
| MNIST-wo7 | train acc. | 1.000 | 0.994 | 0.989 | 0.962 | 0.997 |
|  | test acc. | 0.995 | 0.992 | 0.990 | 0.967 | 0.994 |
|  | NCG score | 0.507 | 0.672 | 0.703 | 0.713 | 0.594 |
| MNIST-wo8 | train acc. | 1.000 | 0.993 | 0.987 | 0.966 | 0.997 |
|  | test acc. | 0.994 | 0.990 | 0.987 | 0.966 | 0.995 |
|  | NCG score | 0.416 | 0.493 | 0.497 | 0.491 | 0.446 |
| MNIST-wo9 | train acc. | 1.000 | 0.996 | 0.992 | 0.962 | 0.997 |
|  | test acc. | 0.996 | 0.994 | 0.992 | 0.964 | 0.995 |
|  | NCG score | 0.577 | 0.714 | 0.691 | 0.703 | 0.660 |

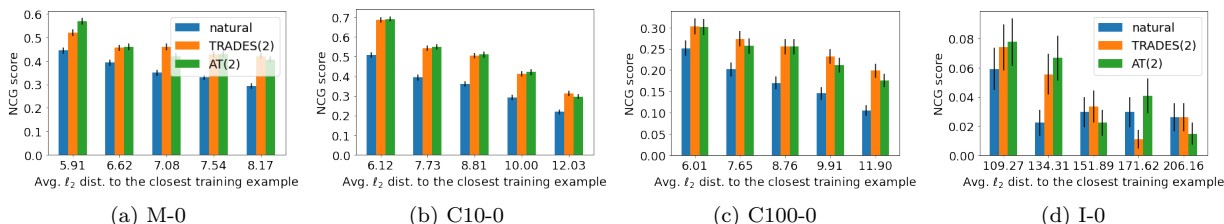

(a) M-0      (b) C10-0      (c) C100-0      (d) I-0

Figure 9: The NCG score of examples from unseen classes and the distance to the closest training example for MNIST, CIFAR10, CIFAR100, and ImageNet-100 in the pixel space.

Table 18: The train, test, and NCG scores of nine different variations of CIFAR10, CIFAR100, and ImgNet100 datasets and five training methods in the pixel space.

|  |  | natural | AT(2) | TRADES(2) | TRADES(4) | TRADES(8) |
|---|---|---|---|---|---|---|
| CIFAR10-wo0 | train acc. | 1.000 | 0.999 | 0.992 | 0.870 | 0.878 |
|  | test acc. | 0.898 | 0.729 | 0.716 | 0.660 | 0.761 |
|  | NCG score | 0.355 | 0.494 | 0.492 | 0.520 | 0.483 |
| CIFAR10-wo4 | train acc. | 1.000 | 1.000 | 0.990 | 0.874 | 0.508 |
|  | test acc. | 0.886 | 0.754 | 0.742 | 0.700 | 0.485 |
|  | NCG score | 0.222 | 0.361 | 0.333 | 0.331 | 0.289 |
| CIFAR10-wo9 | train acc. | 1.000 | 1.000 | 0.992 | 0.948 | 0.778 |
|  | test acc. | 0.885 | 0.725 | 0.712 | 0.732 | 0.641 |
|  | NCG score | 0.145 | 0.212 | 0.192 | 0.247 | 0.245 |
| CIFAR100-wo0 | train acc. | 1.000 | 0.998 | 0.995 | 0.943 | 0.902 |
|  | test acc. | 0.741 | 0.554 | 0.547 | 0.576 | 0.607 |
|  | NCG score | 0.175 | 0.240 | 0.252 | 0.252 | 0.206 |
| CIFAR100-wo4 | train acc. | 1.000 | 0.998 | 0.995 | 0.857 | 0.859 |
|  | test acc. | 0.743 | 0.544 | 0.543 | 0.492 | 0.553 |
|  | NCG score | 0.137 | 0.192 | 0.191 | 0.187 | 0.185 |
| CIFAR100-wo9 | train acc. | 1.000 | 0.996 | 0.995 | 0.950 | 0.527 |
|  | test acc. | 0.727 | 0.547 | 0.537 | 0.585 | 0.431 |
|  | NCG score | 0.222 | 0.353 | 0.412 | 0.427 | 0.465 |
| ImgNet100-wo0 | train acc. | 1.000 | 0.999 | 0.994 | 0.983 | 0.704 |
|  | test acc. | 0.529 | 0.417 | 0.393 | 0.354 | 0.320 |
|  | NCG score | 0.033 | 0.044 | 0.041 | 0.054 | 0.067 |
| ImgNet100-wo1 | train acc. | 1.000 | 0.999 | 0.995 | 0.972 | 0.783 |
|  | test acc. | 0.534 | 0.414 | 0.385 | 0.356 | 0.316 |
|  | NCG score | 0.047 | 0.049 | 0.051 | 0.061 | 0.072 |
| ImgNet100-wo2 | train acc. | 1.000 | 0.999 | 0.994 | 0.971 | 0.695 |
|  | test acc. | 0.537 | 0.394 | 0.388 | 0.353 | 0.320 |
|  | NCG score | 0.027 | 0.028 | 0.033 | 0.044 | 0.049 |

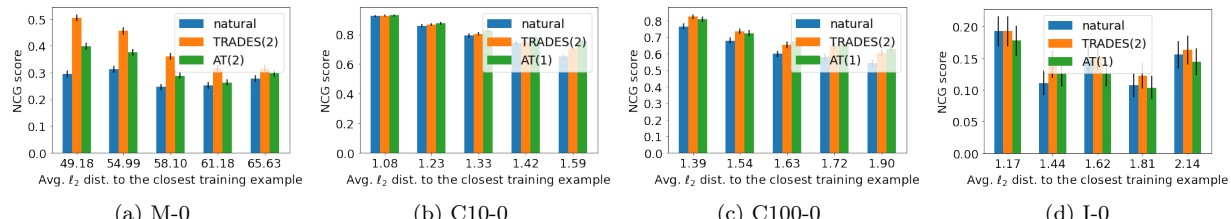

(a) M-0      (b) C10-0      (c) C100-0      (d) I-0

Figure 10: The NCG score of examples from unseen classes and the distance to the closest training example for MNIST, CIFAR10, CIFAR100, and ImageNet-100 in the feature space.

|  |  | natural | AT(2) | TRADES(2) | TRADES(4) | TRADES(8) |
|---|---|---|---|---|---|---|
| MNIST-wo0 | train acc. | 1.00 | 1.00 | 1.00 | 1.00 | 1.00 |
|  | test acc. | 0.99 | 0.99 | 0.99 | 0.99 | 0.99 |
|  | NCG score | 0.28 | 0.32 | 0.39 | 0.49 | 0.55 |
| MNIST-wo1 | train acc. | 1.00 | 1.00 | 1.00 | 1.00 | 1.00 |
|  | test acc. | 0.99 | 0.99 | 0.99 | 0.99 | 0.99 |
|  | NCG score | 0.14 | 0.21 | 0.27 | 0.50 | 0.51 |
| MNIST-wo2 | train acc. | 1.00 | 1.00 | 1.00 | 1.00 | 1.00 |
|  | test acc. | 0.99 | 0.99 | 0.99 | 0.99 | 1.00 |
|  | NCG score | 0.41 | 0.46 | 0.53 | 0.59 | 0.62 |
| MNIST-wo3 | train acc. | 1.00 | 1.00 | 1.00 | 1.00 | 1.00 |
|  | test acc. | 0.99 | 0.99 | 0.99 | 1.00 | 0.99 |
|  | NCG score | 0.68 | 0.71 | 0.73 | 0.73 | 0.74 |
| MNIST-wo4 | train acc. | 1.00 | 1.00 | 1.00 | 1.00 | 1.00 |
|  | test acc. | 0.99 | 0.99 | 0.99 | 1.00 | 1.00 |
|  | NCG score | 0.78 | 0.73 | 0.77 | 0.81 | 0.86 |
| MNIST-wo5 | train acc. | 1.00 | 1.00 | 1.00 | 1.00 | 1.00 |
|  | test acc. | 0.99 | 0.99 | 1.00 | 1.00 | 0.99 |
|  | NCG score | 0.61 | 0.63 | 0.65 | 0.68 | 0.69 |
| MNIST-wo6 | train acc. | 1.00 | 1.00 | 1.00 | 1.00 | 1.00 |
|  | test acc. | 0.99 | 1.00 | 1.00 | 1.00 | 1.00 |
|  | NCG score | 0.54 | 0.58 | 0.60 | 0.65 | 0.66 |
| MNIST-wo7 | train acc. | 1.00 | 1.00 | 1.00 | 1.00 | 1.00 |
|  | test acc. | 0.99 | 0.99 | 1.00 | 1.00 | 1.00 |
|  | NCG score | 0.53 | 0.54 | 0.61 | 0.68 | 0.67 |
| MNIST-wo8 | train acc. | 1.00 | 1.00 | 1.00 | 1.00 | 1.00 |
|  | test acc. | 0.99 | 0.99 | 0.99 | 1.00 | 0.99 |
|  | NCG score | 0.46 | 0.47 | 0.51 | 0.56 | 0.59 |
| MNIST-wo9 | train acc. | 1.00 | 1.00 | 1.00 | 1.00 | 1.00 |
|  | test acc. | 0.99 | 1.00 | 1.00 | 1.00 | 1.00 |
|  | NCG score | 0.61 | 0.71 | 0.71 | 0.73 | 0.79 |

Table 19: The train, test, and NCG scores of 10 MNIST datasets and 5 training methods in the feature space.

Table 20: The train, test, and NCG scores of nine different variations of CIFAR10, CIFAR100, and ImgNet100 datasets and five training methods in the feature space. We use different radius for AT since not all converge well when the radius is large ($r = 2$) For CIFAR10 and CIFAR100, we use AT(1); for ImgNet100, we use AT(.5).

|  |  | natural | AT(.5)/(1) | TRADES(2) | TRADES(4) | TRADES(8) |
|---|---|---|---|---|---|---|
| CIFAR10-wo0 | train acc. | 1.00 | 1.00 | 1.00 | 1.00 | 1.00 |
|  | test acc. | 0.89 | 0.89 | 0.89 | 0.90 | 0.90 |
|  | NCG score | 0.80 | 0.83 | 0.81 | 0.83 | 0.83 |
| CIFAR10-wo4 | train acc. | 1.00 | 1.00 | 1.00 | 1.00 | 1.00 |
|  | test acc. | 0.88 | 0.88 | 0.88 | 0.89 | 0.88 |
|  | NCG score | 0.82 | 0.84 | 0.82 | 0.85 | 0.85 |
| CIFAR10-wo9 | train acc. | 1.00 | 1.00 | 1.00 | 1.00 | 1.00 |
|  | test acc. | 0.88 | 0.88 | 0.88 | 0.89 | 0.89 |
|  | NCG score | 0.84 | 0.89 | 0.83 | 0.88 | 0.87 |
| CIFAR100-wo0 | train acc. | 1.00 | 1.00 | 1.00 | 1.00 | 1.00 |
|  | test acc. | 0.72 | 0.73 | 0.73 | 0.74 | 0.74 |
|  | NCG score | 0.63 | 0.70 | 0.69 | 0.68 | 0.68 |
| CIFAR100-wo4 | train acc. | 1.00 | 1.00 | 1.00 | 1.00 | 1.00 |
|  | test acc. | 0.72 | 0.73 | 0.73 | 0.74 | 0.74 |
|  | NCG score | 0.69 | 0.74 | 0.75 | 0.73 | 0.74 |
| CIFAR100-wo9 | train acc. | 1.00 | 1.00 | 1.00 | 1.00 | 1.00 |
|  | test acc. | 0.70 | 0.72 | 0.72 | 0.73 | 0.73 |
|  | NCG score | 0.66 | 0.74 | 0.72 | 0.71 | 0.71 |
| ImgNet100-wo0 | train acc. | 0.99 | 0.57 | 0.33 | 0.98 | 0.98 |
|  | test acc. | 0.22 | 0.25 | 0.26 | 0.26 | 0.26 |
|  | NCG score | 0.11 | 0.16 | 0.15 | 0.12 | 0.13 |
| ImgNet100-wo1 | train acc. | 1.00 | 0.56 | 0.32 | 0.98 | 0.98 |
|  | test acc. | 0.22 | 0.24 | 0.27 | 0.26 | 0.25 |
|  | NCG score | 0.13 | 0.15 | 0.18 | 0.14 | 0.15 |
| ImgNet100-wo2 | train acc. | 1.00 | 0.60 | 0.33 | 0.98 | 0.98 |
|  | test acc. | 0.22 | 0.25 | 0.26 | 0.26 | 0.26 |
|  | NCG score | 0.11 | 0.15 | 0.15 | 0.14 | 0.14 |

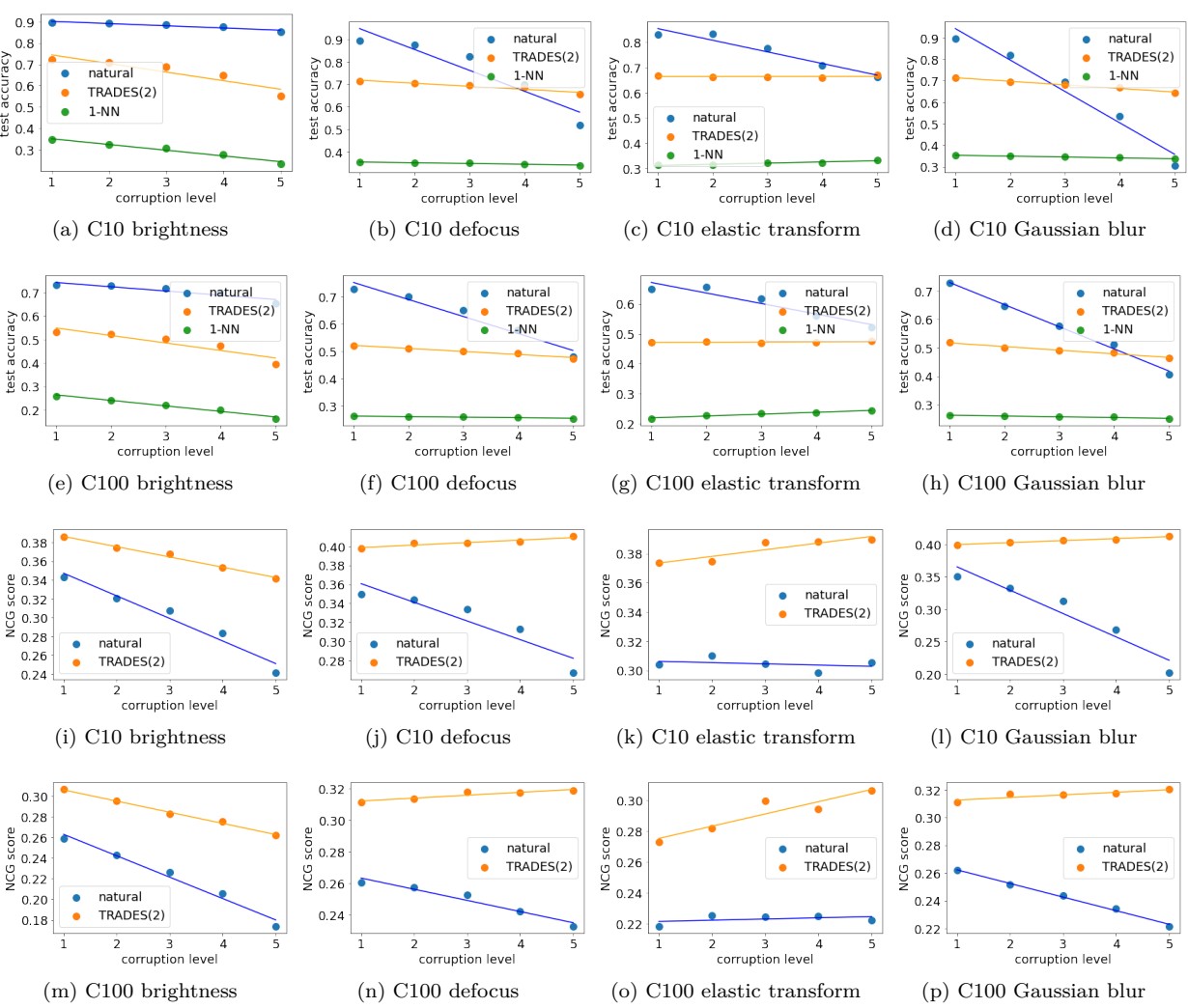

Figure 11: The slopes of the test accuracy of naturally trained models and TRADES(2) on CIFAR10 in the pixel space.

