# OpenReview forum: "Probing Predictions on OOD Images via Nearest Categories"
_TMLR — Accepted by TMLR_

### Review · Reviewer_eC2G · 2022-11-19

**Summary Of Contributions:**

This paper points out that robust neural networks (such as ones trained with adversarial training and TRADES) exhibit better out-of-distribution generalization in the sense that their predictions align well with 1-nearest neighbor labels. This behavior motivates a new metric called NCG to measure OOD generalization. The authors simulate two kinds of OOD data: unseen classes and corrupted data. The robust networks admit better NCG accuracy when evaluated with these two OOD data. It is then discussed that robust networks generalize well on OOD data because their local smoothness would be extended well to OOD data. The proposed metric could be used as a cheap proxy for OOD detection.

**Audience:**

Yes

**Broader Impact Concerns:**

Not applicable.

**Claims And Evidence:**

Yes

**Requested Changes:**

Major requests are presented in the above "weaknesses".
Below, I leave a few minor ones.

- In the last paragraph of Section 1.2, the authors argue "the NCG behavior is a consequence of the inductive bias produced by neural networks," which seems to lack further discussion. What kind of inductive biases do you suppose in this context?
- In the first line of Section 3, does WRN stand for Wide-ResNet?
- In the second line of Section 4.1, I do not see a clear meaning of "NCG correct data" and "NCG incorrect data". May I ask the authors to provide their formal meanings? Given that, the results in Section 4.1 can be better understood.


**Strengths And Weaknesses:**

## Strengths
1. Significant experimental evaluations: The authors provide experiments with two types of OOD data with standard adversarial training methods. Remarkably, they provide statistical tests to support the validity of the significance of the experimental results to conclude NCG accuracy is higher for robust networks.
2. The Voronoi tessellation perspective of robust training: After Section 3.2, the authors discuss how different robustness region in robust training influences the resulting performance and consider the sub-Voronoi region, which seems to be a reasonable choice. This robustness region may not have been discussed so far partially because the computation is challenging. The authors discuss some approximation methods, which are appealing.

## Weaknesses
1. Motivation is slightly unclear: The main result of this paper is to reveal a correlation between network robustness and NCG accuracy. However, I am not fully motivated because OOD data may contain unseen classes in the training phase and thus 1-NN accuracy occasionally disregards those unseen classes because 1-NN classifies any input to a known class label. Despite that the correlation seems interesting, it is unclear how NCG accuracy is helpful as an OOD detection measurement.
2. Discussion in Section 3.1 would be a little bit confusing: In Section 3.1, the authors discuss why robust models have higher NGC accuracy and suppose that robust models may have better local smoothness and hence the predictions are locally consistent. Hence, I suppose that the authors would like to claim that OOD data tend to fall in the local balls of the training data and they are classified correctly. However, the experiments in the latter part of this section reveal that OOD data are consistently farther than the worst-case adversarial examples surrounding the training data. Though this result is somewhat intuitive, it does not seem to support higher NCG accuracy for OOD data. In other words, how the local smoothness behavior is extended to the global behavior still remains unclear. Can the authors give a little bit more discussion in this regard?
3. The illustrative example in Figure 3 would be misleading: The illustration in Figure 3(d) seems to support the superiority of the sub-Voronoi region in the sense that we may obtain a better boundary without crossing over OOD data using this robustness region. However, what if the yellow OOD cluster is located at a slightly more right region? The sub-Voronoi region seems to induce a boundary that crosses such OOD data. That being said, the goodness of the robustness region would depend on the nature of OOD data significantly. This would be one of the reasons why several robustness regions compared in Table 3 are comparable to each other. This perspective could be further discussed.

---

> ### Author Response · Authors · 2022-12-10
> **Response (1/2)**
>
> - _Motivation is slightly unclear: The main result of this paper is to reveal a correlation between network robustness and NCG accuracy. However, I am not fully motivated because OOD data may contain unseen classes in the training phase and thus 1-NN accuracy occasionally disregards those unseen classes because 1-NN classifies any input to a known class label. Despite that the correlation seems interesting, it is unclear how NCG accuracy is helpful as an OOD detection measurement._
>
> The main objective of this paper is to understand an OOD generalization property (or how a neural network makes predictions on OOD data) of neural networks. We consider nearest category generalization (NCG) to measure how often networks label an OOD example as the label of its nearest training example. We would like to clarify that we do not intend to use NCG accuracy as an OOD detection measurement. Instead, we use NCG as a probing measurement to measure the nearest category generalization property of neural networks under different datasets and training methods. As you point out, 1-NN labels all points, so it is feasible to use it for both in- and out-of-distribution data.
>
> As for the connection between NCG and OOD detection, please refer to the general comment.
>
> - _Discussion in Section 3.1 would be a little bit confusing: In Section 3.1, the authors discuss why robust models have higher NGC accuracy and suppose that robust models may have better local smoothness and hence the predictions are locally consistent. Hence, I suppose that the authors would like to claim that OOD data tend to fall in the local balls of the training data and they are classified correctly. However, the experiments in the latter part of this section reveal that OOD data are consistently farther than the worst-case adversarial examples surrounding the training data. Though this result is somewhat intuitive, it does not seem to support higher NCG accuracy for OOD data. In other words, how the local smoothness behavior is extended to the global behavior still remains unclear. Can the authors give a little bit more discussion in this regard?_
>
> You are actually pointing out what we find most interesting, and surprising, about our results. In Section 3.1, our main objective is to present evidence to *reject* the hypothesis that the increase in NCG accuracy through robust training is caused by covering the OOD examples within the robust radius. We do so by showing that OOD examples are consistently further away. This experiment implies that although adversarial training does increase NCG accuracy, it cannot simply be explained by the larger robust radius. This leads us to believe that the inductive bias from the neural network (architecture, optimizer, and cross-entropy-based loss function) seems to have a larger impact on the NCG property.
>
> - _The illustrative example in Figure 3 would be misleading: The illustration in Figure 3(d) seems to support the superiority of the sub-Voronoi region in the sense that we may obtain a better boundary without crossing over OOD data using this robustness region. However, what if the yellow OOD cluster is located at a slightly more right region? The sub-Voronoi region seems to induce a boundary that crosses such OOD data. That being said, the goodness of the robustness region would depend on the nature of OOD data significantly. This would be one of the reasons why several robustness regions compared in Table 3 are comparable to each other. This perspective could be further discussed._
>
> We agree with the reviewer, and this is a good point. For any “good” decision boundary, there will be “bad” distributions of OOD data. So perhaps you are correct that the OOD data for real experiments are “bad” in the sense that none of the decision boundaries properly captures them.
>
> Our motivation, however, is not to find the most robust classifier, but to see how to increase NCG accuracy through a new training method. NCG accuracy basically measures how well the decision boundary of a neural network matches the decision boundary of a 1-nearest neighbor classifier on OOD examples. Thus, the sub-Voronoi region formed by the training examples would be a natural choice of the robust region (this is because if a classifier has a decision boundary that matches the sub-Voronoi region, and the label of each region is the same as the training example in that region, then the classifier would have a 100% NCG accuracy). The idea of this choice is that if an OOD example is given with no assumption made on these given examples. Then the best thing we can do is to predict this given example with the same label as the closest training example.

---

> ### Author Response · Authors · 2022-12-10
> **Response (2/2)**
>
> - _As we can see in Figure 3(d), the neural network trained on the toy dataset has two vertical boundaries near x=1 and x=6 as its decision boundary, which is close to the boundary of the 1-nearest neighbor classifier trained on the same dataset. This shows that, at least in low dimensional data, training on sub-Voronoi regions could indeed improve NCG accuracy. And in the case where the yellow OOD cluster is located slightly more right (passes the x-6 line), then some of these points will have an orange point as their nearest neighbor. This means some of the yellow points would need to be predicted as the orange label in order to be considered as “NCG correct” (see the definition at the end of this response)._
>
> However, when training on higher dimensional data, as in Table 3, things seem to be working differently. This could be caused by the nature of harder robust optimization in high dimensional space (similar to adversarial robustness is still hard) or other reasons. But we conclude that the inductive bias of the neural network has a larger impact on the NCG accuracy than the robust region the network is trained on. Also, the dataset has a big impact on the easiness of the problem (For example, natural training on M-4 has 0.76 NCG accuracy and the best NCG accuracy on M-0 is only 0.51).
>
> - _In the last paragraph of Section 1.2, the authors argue "the NCG behavior is a consequence of the inductive bias produced by neural networks," which seems to lack further discussion. What kind of inductive biases do you suppose in this context?_
>
> Inductive bias refers to the biases that arise because of the way a model is trained, which includes optimization algorithm, learning objective, network architecture, and dataset. What we mean in this context is that NCG is one of the inductive biases produced by these factors.
>
> - _In the first line of Section 3, does WRN stand for Wide-ResNet?_
>
> Thanks for pointing this out. Yes, WRN-40-10 stands for Wide-ResNet with the depth set to 40 and the widen factor set to 10. We will make sure this is clear in the revision.
>
> - _In the second line of Section 4.1, I do not see a clear meaning of "NCG correct data" and "NCG incorrect data". May I ask the authors to provide their formal meanings? Given that, the results in Section 4.1 can be better understood._
>
> If an example $x$ is NCG correct, it means that the prediction on $x$ is the same as the label of the closest training example to $x$. If the prediction on $x$ is different from the label of the closest training example to $x$, $x$ is considered NCG incorrect. So, NCG accuracy measures how many OOD examples have the same prediction from a neural network and a 1-nearest neighbor classifier.

---

> > ### Comment · Reviewer_eC2G · 2022-12-12
> > **Thanks for the responses**
> >
> > Thanks for dealing with my comments dedicatedly.
> >
> > As for the connection between OOD and NCG, please see [my discussion above](https://openreview.net/forum?id=fTNorIvVXG&noteId=lBQtuqGf0c).
> >
> > On *The illustrative example in Figure 3*: In my opinion, the figure is a good illustration of the new training scheme. Nevertheless, this may cause a confusion for readers. In this regard, I recommend to add some discussions like what you said
> >
> > > For any “good” decision boundary, there will be “bad” distributions of OOD data.
> >
> > On *inductive bias*: I understand the claim "What we mean in this context is that NCG is one of the inductive biases produced by these factors.", but what I do not understand now is **what kind of the underlying mechanism** (or in other words, what inductive bias) causes higher NCG. As you mentioned, inductive bias subsumes a lot of factors such as initialization scale, learning rate choice and scheduling, optimizers, and model architectures. Oftentimes these factors may jointly affect. Even so, it would be a nice idea to discuss this perspective in a little bit more detail.

---

### Review · Reviewer_TdWQ · 2022-12-01

**Summary Of Contributions:**

This paper studies the behavior of a neural network optimized with adversarially robust training on out-of-ditribution (OOD) data. The authors empirically investigate how often the prediction given by a robust network and the label of the nearest training data agree with each other. This frequency of agreement is referred to as Nearest Category Generalization (NCG) accuracy. The results suggest that the NCG accuracy is consistently beyond the chance level, robust networks have much higher NCG accuracy than ordinarily trained networks, and in the context of prediction on corrupted examples, the NCG accuracy and the prediction accuracy have positive correlation. The paper also provides a theoretical analysis, which says that NCG is much easier than OOD detection in terms of sample complexity.

**Audience:**

Yes

**Broader Impact Concerns:**

I do not have any concern with this paper about broader impact.

**Claims And Evidence:**

Yes

**Requested Changes:**

- Regarding the second weakness in the Strengths And Weaknesses section, it would be nice if the authors could show some classifier(s) other than the 1-nearest-neighbor rule do not have the property observed for NCG. That would indicate that the metric based on NCG is special in OOD prediction.
- In the paragraph about "Distance Metrics for NCG," the paper mentions, "we believe the $\ell_\infty$ results would be similar." I wonder why we would be particularly interested in $\ell_\infty$. Also, what makes the authors believe so?
- In the setup of Section 3, is the scale of the robustness radii (0.5, 1, 2, 4, 8) small enough?
- In Table 1, what does the row AT(2)/(1)/(.5) refer to?
- In Table 1 and 3, the scores for "natural training" seem very different. Why?

**Strengths And Weaknesses:**

### Strengths
- The paper is well-written and easy to follow.
- Behavior of robust networks in extrapolation is an interesting and important topic to be investigated.
- The use of the NCG accuracy for studying OOD prediction is novel.
- The series of experiments are well-designed and insightful.
- The connection between NCG and OOD is somewhat surprising and interesting.

### Weaknesses
- The motivation for studying the NCG accuracy seems weak. I don't see a natural connection between NCG and OOD.
- There might be a gap from the empirical results to the conclusion that the NCG accuracy is useful for knowing the OOD prediction performance. NCG accuracy positively correlates with OOD prediction accuracy possibly because good classifiers tend to agree with other classifiers. It might be the case that any classifier other than the 1-nearest-neighbor rule, with reasonably high accuracy, has the same property. The presented experiments do not deny this possibility.
- The connection between NCG and OOD seems to have little practical implications.

---

> ### Author Response · Authors · 2022-12-10
> **Response**
>
> For the connection between NCG and OOD detection, please refer to the general comment.
>
> - _Regarding the second weakness in the Strengths And Weaknesses section, it would be nice if the authors could show some classifier(s) other than the 1-nearest-neighbor rule do not have the property observed for NCG. That would indicate that the metric based on NCG is special in OOD prediction._
>
> We would like to clarify that we don’t think the 1-nearest-neighbor classifier is the only classifier that could give us interesting insight into OOD prediction. However, we would like to say that NCG is somehow special for image datasets. For the choice of the classifier to use, one wants to choose a classifier that is easy to reason about so that when the decision boundary of the network matches the classifier, we know its meaning. This leaves us just only a handful of options – linear classifiers, decision trees, and nearest neighbors. Among them, the linear classifier and decision tree requires each dimension of the feature to be meaningful, which is not the case for image data as interpreting each pixel independently contains too little information and each feature in the feature space doesn't have a concrete meaning. This makes 1 nearest neighbor classifier the most natural choice.
>
> - _In the paragraph about "Distance Metrics for NCG," the paper mentions, "we believe the \ell_\infty results would be similar." I wonder why we would be particularly interested in \ell_\infty. Also, what makes the authors believe so?_
>
> A lot of robustness work focuses on \ell_\infty as an image similarity metric because when two images are very close in \ell_\infty distance then they can be considered imperceptibly different to humans. The reason is that all the pixels must be close in value. On the other hand, being close in \ell_2 distance does not exclude the possibility of having some pixels be very different, which would be noticeable. Neither metric is perfect for aligning with human perception.
>
> - _In the setup of Section 3, is the scale of the robustness radii (0.5, 1, 2, 4, 8) small enough?_
>
> We chose these radii because they are much less than the minimum distance between differently labeled images in \ell_2 distance in pixel space. It is harder to choose an appropriate radius for embedding spaces. In other words, the robust radii are small compared to the distance from an OOD example to the closest training example; according to Figure 2, the radii are a lot smaller. The point is that enforcing smoothness in a small radius should not, a priori, dictate the network’s prediction on “far away” OOD examples, so this behavior is surprising. If you have another criteria for small/large enough radii, please let us know.
>
> - _In Table 1, what does the row AT(2)/(1)/(.5) refer to?_
>
> It refers to adversarial training (AT) with robust radius r=2 for MNIST, r=1 for CIFAR10/CIFAR100, and r=0.5 for ImgNet100. When performing adversarial training in the feature space, too large of an r can make the model underfit too much, thus, we set different r for different dataset.
>
> -_In Table 1 and 3, the scores for "natural training" seem very different. Why?_
>
> Table 1 shows the average over all held-out classes while Table 3 shows the NCG accuracy for the results on each individual held-out class. Take MNIST as an example. To achieve the 0.49 for natural training in Table 1, you would take the average and standard error over the “natural” column in Table 15 of the appendix. For the numbers in Table 3, we only show some rows in Table 15.

---

### Review · Reviewer_Svr8 · 2022-12-02

**Summary Of Contributions:**

This paper focuses on understanding the behaviour of computer vision models on out-of-distribution (OOD) image data. To this end, the authors propose the Nearest Category Generalization or NCG accuracy metric that measures the match between the prediction of the model and the 1-nearest-neighbor classifier, where the 1-nearest-neighbor is computed using the $\ell_2$ metric in the input pixel space or some feature space generated by an encoder. This metric appears to implicitly evaluate the "smoothness" of the model, with more smooth models having higher NCG accuracy.

Using this metric and two kinds of OOD image data -- images from unseen classes and corrupted images -- the author empirically demonstrate various properties of different models. The experimental results show that robustly trained models have higher NCG accuracy than naturally trained models, and locally constrained robust training has larger global effect on the smoothness of the model than what the local constraint imposes. For OOD data involving corrupted images, the authors interpret the empirical results to claim that higher NCG accuracy can imply higher test accuracy.

Based on the insights from the experiments, the authors also propose a new form of locally constrained robust training that adapts the size of the "local region" based on the geometry of the data, and provide various practical approximations to this form of locally constrained robust training.



**Audience:**

Yes

**Broader Impact Concerns:**

There are not ethical implications of this work to the best of my understanding.

**Claims And Evidence:**

No

**Requested Changes:**

To me, addressing weaknesses 1, 2 and 3 are critical for securing my recommendation, with the strong suggestion that addressing weaknesses 4-5 will improve the paper even further and highlight the contributions.


**Strengths And Weaknesses:**

### Strengths

1. Understanding the OOD behaviour of a model is an extremely problem and the attempt in this paper to quantify a form of OOD behaviour with the NCG accuracy metric can be very useful.

2. The paper empirically evaluates the NCG accuracy to gain insights regarding the OOD behaviour of different models. For this purpose, the authors consider a reasonable combination of datasets, models and robust training procedures.

3. Highlighting the global effect on the smoothness of the models trained with local smoothness constraints can be very critical. While it seems like the effect on the global behaviour is usually positive, it is important to understand and quantify this global effect emerging from local constraints since that is an implicit (and at times unintended) effect, and hence can be problematic in some problems and applications.

4. Based on the insights from the experiments, the sub-Voronoi region based local smoothing appears to be a very well-motivated robust training scheme. It seems natural that the size of the local region in the local smoothing constraints should depend on the geometry of dataset and be adaptive instead of being uniform across all training points in the robust training (as in commonly used in adversarial training).


### Weaknesses

1. One of the main weaknesses of this paper is the lack of clear connection between the considered NCG accuracy and OOD behaviour. NCG accuracy of a model is related to how smooth a model in all directions. While smoothness is interesting, I do not get a good motivation or intuition as to why NCG accuracy is a good quantification of OOD behaviour. Without this clear connection/motivation/intuition, it is not clear what "better-than-random" NCG accuracy of all models or "improved" NCG accuracy of robustly trained models (over naturally trained models) mean. NGC accuracy implicitly quantifies smoothness of the model but that does not have any direct relationship to OOD behaviour. If there is any connection between NCG accuracy and OOD behaviour (positive or negative), what would it mean for the 1-nearest-neighbor classifier which has a NCG accuracy of 100%?


2. The motivation seems more unclear to me with the discussion in Section 5 regarding "NCG is an easier problem in theory than OOD detection". This discussion is not well introduced or positioned in my opinion. First it is not clear what the "problem of NCG" is. Is it the problem of increasing NCG accuracy or the problem of computing NCG accuracy? From Theorem 1, NCG seems to mean "classifying a sample with its 1-nearest-neighbor label" but that again is not clear in its meaning. It is also not clear if these two problems (NCG vs OOD detection) can even be compared. Increasing NCG accuracy implies creating a model than matches the 1-nearest-neighbor classifier by potentially increasing smoothness of the model is all directions, and this has nothing to do with the properties of the OOD data in any way. OOD detection on the other hand relates to how similar or different the distributions of the in-distribution training data and OOD data are, with OOD detection becoming easier as the OOD data starts becoming more and more different than the in-distribution.

3. For the empirical evaluation with corrupted OOD images, the claim that higher NCG accuracy implies higher test accuracy appears problematic or at least incomplete. The claim that higher NCG accuracy with corruption-based OOD implies higher test accuracy seems to imply that there is an underlying assumption that the 1-nearest-neighbor label in the pixel/feature space is the right label for the corrupted image. Why is that necessarily true? This needs better motivation, and this assumption is orthogonal to the model, and the robust training procedure. If the corruption was large enough to change the class of the 1-nearest-neighbor (in the pixel/feature space), then there would be no clear correlation between the NGC accuracy and the test accuracy.


4. It is not clear whether the (approximate) adaptive sub-Voronoi-based locally constrained robust training schemes are a novel contribution. If it is so, it is not clear why are they excluded from the evaluation with corruption-based OOD.


5. Finally, the precise contributions of this paper should be clarified better. Right now, it appears that the main contribution is this NCG accuracy metric and its evaluation for various models with some insights. Another partial contribution seems to be the adaptive sub-Voronoi based locally constrained robust training scheme, but this scheme is not thoroughly evaluated, and the results of the partial evaluation do not give a clear picture of its advantage. Even if the NCG accuracy metric is meaningful (which, to me, is currently in question), these contributions appear to be somewhat limited.



### Questions, Concerns and Comments


- Are all global effects of local smoothing positive or neutral or can there be negative global effects?

- Why should we focus on the 1-nearest-neighbour classifier with $\ell_2$ distance in pixel/feature space as the base model?

- Regarding "the behavior on OOD data is far from random", why should we expect the OOD behavior of any trained model to be random?

- Figure 1 is not very clear. How is it related to OOD generalization? The smoothness of the model might be different in different directions (especially in high dimensions), but how does that connect with OOD generalization? The learned model has increased or decreased smoothness is different direction based on the decision boundary it is learning. Moreover, It might be useful to consider the directions where the smoothness increases after robust training (relative to natural training).

- From the empirical results, the authors seem to conclude that on OOD data, the models behave somewhat like a 1-nearest-neighbor classifier. The NCG accuracies are better than random but the raw NGC accuracies are often much too less to say that the model behaves like the 1-nearest-neighbour classifier unless I misunderstand the discussion. Furthermore, the improvement of robust training over natural training (in terms of NCG accuracy) is statistically significant, but the actual increases are still fairly small -- this might imply that the global behaviour is changing but not that drastically since the NCG accuracies of both the natural and robust training would be considered quite low. The results in Figure 6 seem to convey a similar message that the models continue to remain locally smooth -- the NCG accuracy drops as the distance increases. So there is some global effect but it does not appear to be significant.

- Doesn't the difference between pixel space and feature space results boil down to whether we are enforcing smoothness across the end-to-end model or just across the final few layers of the model? In the latter case, it is not surprising that robust training is close to natural since the smoothness enforcement is only on part of the model and hence the effect is somewhat limited.

- Where is figure 4 referenced? It seems to appear out of the blue and not discussed anywhere in the main paper. However, I might have missed something in the text.

- The theorem statement appears incomplete with respect to the role of $\epsilon$ and $C$ beyond just universal constants -- the sample complexity deteriorates with smaller $\epsilon$ but it is not clear what tradeoff is present and why one should not just pick $\epsilon \approx 1/2$ and get a better sample complexity than that of $O(C \log C)$. I understand that $\epsilon$ pertains to the distribution of the data. But the role of $\epsilon$ needs to be presented rigorously in the theorem statement.

- Why should one consider corrupted data (used in the experiments) OOD especially if the corruption does not change the class of the nearest-neighbor (in pixel/feature space)?

---

> ### Author Response · Authors · 2022-12-10
> **Response (1/4)**
>
> - _One of the main weaknesses of this paper is the lack of clear connection between the considered NCG accuracy and OOD behaviour. NCG accuracy of a model is related to how smooth a model in all directions. While smoothness is interesting, I do not get a good motivation or intuition as to why NCG accuracy is a good quantification of OOD behaviour. Without this clear connection/motivation/intuition, it is not clear what "better-than-random" NCG accuracy of all models or "improved" NCG accuracy of robustly trained models (over naturally trained models) mean. NGC accuracy implicitly quantifies smoothness of the model but that does not have any direct relationship to OOD behaviour. If there is any connection between NCG accuracy and OOD behaviour (positive or negative), what would it mean for the 1-nearest-neighbor classifier which has a NCG accuracy of 100%?_
>
> Please see the connection between NCG accuracy and OOD behavior in the general comment.
>
> We would like to clarify that NCG accuracy on OOD data does not imply smoothness in all directions. Instead, adversarial robustness measures the smoothness in all directions, while NCG accuracy on OOD data measures smoothness in a certain natural image direction (since all OOD data used in this study are natural images). OOD data can be very diverse (anything not sampled from the original distribution is considered OOD); thus, OOD behaviors can also be diverse. NCG accuracy only characterizes one kind of behavior.
>
> NCG accuracy is a way to probe what the decision boundary of a model looks like on the support of the OOD data on which the NCG accuracy is computed. In terms of why NCG accuracy is a good quantification of OOD behavior, there are four reasons:
> - It can be used to uncover a previously unknown generalization property of neural networks, which is that neural networks tend to predict OOD examples as the label of the closest training example. This finding is supported by the “better-than-random” NCG accuracy result.
> - We also find that the NCG behavior is not caused by extending the robust region around each training example uniformly. But rather, it is caused by extending the robust region towards a certain natural image direction through the inductive bias of the neural network itself (enforcing on different robust regions could help but only to a certain extent).
> - It is correlated with the test accuracy on corrupted data, which implies that the inductive biases needed for high NCG accuracy may be similar to the inductive biases needed for high test accuracy on corrupted data. This can help to understand how neural networks generalize on OOD data.
> - Finally, in Section 5, we show how NCG accuracy may provide us with a measurement of robustness when OOD detection is nearly impossible.
>
> We would like to clarify that the “improved” NCG accuracy does not mean that we have a good metric, and we want it to be higher. Instead, we want to figure out what kind of loss function produces a model that follows NCG more strictly.
>
> If a model or network has a high NCG accuracy, it suggests that the model has a decision boundary close to the Voronoi diagram (at least on the support of the test data), which will be surprising if neural networks behave this way.
>
> - _The motivation seems more unclear to me with the discussion in Section 5 regarding "NCG is an easier problem in theory than OOD detection". This discussion is not well introduced or positioned in my opinion. First it is not clear what the "problem of NCG" is. Is it the problem of increasing NCG accuracy or the problem of computing NCG accuracy? From Theorem 1, NCG seems to mean "classifying a sample with its 1-nearest-neighbor label" but that again is not clear in its meaning. It is also not clear if these two problems (NCG vs OOD detection) can even be compared. Increasing NCG accuracy implies creating a model than matches the 1-nearest-neighbor classifier by potentially increasing smoothness of the model is all directions, and this has nothing to do with the properties of the OOD data in any way. OOD detection on the other hand relates to how similar or different the distributions of the in-distribution training data and OOD data are, with OOD detection becoming easier as the OOD data starts becoming more and more different than the in-distribution._
>
> NCG problem definition: Want to maximize the following accuracy, which has two components. On an in-distribution example, output its label. On an OOD example, output the 1-NN label. Given a data distribution over both in- and out-of-distribution examples, we can measure the accuracy of an algorithm in this way.
>
> We say that NCG is an easier problem in theory than OOD detection because OOD detection implies good NCG accuracy on OOD examples. If we had a perfect OOD detector, then we could detect the OOD examples and label them with the 1-NN classifier. For the in-distribution examples, we would use any high-accuracy model (like a neural network).

---

> > ### Comment · Reviewer_Svr8 · 2022-12-13
> > **Regarding increased smoothness in "natural directions"**
> >
> > > We would like to clarify that NCG accuracy on OOD data does not imply smoothness in all directions. Instead, adversarial robustness measures the smoothness in all directions, while NCG accuracy on OOD data measures smoothness in a certain natural image direction (since all OOD data used in this study are natural images). OOD data can be very diverse (anything not sampled from the original distribution is considered OOD); thus, OOD behaviors can also be diverse. NCG accuracy only characterizes one kind of behavior.
> >
> > On the discussion on increased smoothness/NGC accuracy in "natural directions", would we not need to show that the NCG accuracy **does not** improve in all directions and just specific directions? If we consider random artificial images (images of appropriate sizes but with random pixel values), the above insight implies that the NCG accuracy **should not** improve on these artificial OOD images. Is that something that is already considered in the paper? Or is such an evaluation not necessary for the point being made in the paper.

---

> > > ### Author Response · Authors · 2022-12-18
> > > **Thank you for responding**
> > >
> > > The adversarial robust radius measures how far the decision boundary extends **in all directions**. Let the robust radius of an example $x$ be $r$, it means that there exist an $x’$ where $\|x-x’\|=r$ and $f(x) \neq f(x’)$. This indicates that the decision boundary does not extend from $x$ to beyond $x’$ in some directions. Thus, for example $x$, the decision boundary is only smooth **in all directions** up to distance $r$.
> > >
> > > In Figure 2, we measure the adversarial robust radius and the distance for each NCG correct OOD example to its closest training example. The latter measures how far the decision boundary extends towards the direction where OOD data lies. Our result suggests that the decision boundary extends toward some natural image directions. However, we cannot say that natural image directions are the **only** directions the decision boundary extends. Thus, random pixel values may or may not follow NCG, depending on the underlying distribution that each pixel is sampled from.

---

> > > > ### Comment · Reviewer_Svr8 · 2022-12-29
> > > > **Smoothness along natural directions and beyond**
> > > >
> > > > Thank you for your response. If I understand correctly, what is being shown is that the adversarially trained models are smoother along certain "natural directions" (although I am not sure how we can define such a "natural direction") compared to models obtained via standard training, but the models can be smooth in other directions as well. I am not sure how to interpret this. The claim in some parts of the paper (such as the illustrative example in Figure 1) is that the model is smoother in **different directions to different extents** contrary to the constraint imposed in adversarial training where the model is constrained to be equally smooth in all directions upto a distance of $r$. Then the empirical evidence in the paper does not appear to satisfy the claim. What am I misinterpreting?

---

> > > > > ### Author Response · Authors · 2022-12-31
> > > > > **Response to "Smoothness along natural directions and beyond"**
> > > > >
> > > > > Thank you for responding!
> > > > >
> > > > > > Thank you for your response. If I understand correctly, what is being shown is that the adversarially trained models are smoother along certain "natural directions" (although I am not sure how we can define such a "natural direction") compared to models obtained via standard training, but the models can be smooth in other directions as well. I am not sure how to interpret this.
> > > > >
> > > > > By natural direction, we are referring to some directions starting from some training examples that point towards some natural images instead of random or adversarial images. For example, to a model trained on MNIST 0-8, directions from images of 0-8 to images of 9 are natural directions.
> > > > >
> > > > > It is correct that we can only claim that the model is smoother in **some** natural directions (on the OOD data we tested on). We cannot claim that the model is smoother **only** in these directions (it is typically infeasible to make such a claim in high dimensional space).
> > > > >
> > > > > > The claim in some parts of the paper (such as the illustrative example in Figure 1) is that the model is smoother in different directions to different extents contrary to the constraint imposed in adversarial training where the model is constrained to be equally smooth in all directions upto a distance of . Then the empirical evidence in the paper does not appear to satisfy the claim. What am I misinterpreting?
> > > > >
> > > > > In our paper, we find that models remain smooth further in the natural directions than the empirical robust radius. Assuming that the empirical robust radius for a point $x$ is $r$, this means that for this point, there exists a direction where the smoothness extends less than $r$. In Figure 2, the blue bars shows the empirical robust radius for each point, and the orange bars show how far away the model remain smooth in the natural directions. We see that the blue bars are far smaller than the orange bars on the x-axis for almost all points. This indicates that there exists a direction in the smoothness that does not extend beyond the natural direction, suggesting that the natural directions are generally much smoother. Thus, we believe our evidence is able to support our claim. Please let us know if this clarifies our claim and if there are any further questions.

---

> ### Author Response · Authors · 2022-12-10
> **Response (2/4)**
>
> - _For the empirical evaluation with corrupted OOD images, the claim that higher NCG accuracy implies higher test accuracy appears problematic or at least incomplete. The claim that higher NCG accuracy with corruption-based OOD implies higher test accuracy seems to imply that there is an underlying assumption that the 1-nearest-neighbor label in the pixel/feature space is the right label for the corrupted image. Why is that necessarily true? This needs better motivation, and this assumption is orthogonal to the model, and the robust training procedure. If the corruption was large enough to change the class of the 1-nearest-neighbor (in the pixel/feature space), then there would be no clear correlation between the NGC accuracy and the test accuracy._
>
> We would like to clarify that we did not claim that higher NCG accuracy "implies" higher test accuracy. Instead, we claim that we found a positive correlation between NCG accuracy and test accuracy on multiple datasets. This result implies that the inductive bias that allows a model to have higher NCG accuracy may overlap with the inductive bias that is required to have high test accuracy on corrupted examples.
> It is not clear whether the (approximate) adaptive sub-Voronoi-based locally constrained robust training schemes are a novel contribution. If it is so, it is not clear why are they excluded from the evaluation with corruption-based OOD.
>
> The algorithm for enforcing robustness on the sub-Voronoi region is new. However, these algorithms tend to underfit and have lower test accuracies. Thus, they do not do well under the corruption-based OOD setting. In addition, the main purpose of these algorithms is to understand how enforcing robustness on different regions can change a model’s NCG behavior. These algorithms successfully served their purpose. To make these algorithms useful, further research into new optimization methods may be needed. It can be an interesting future direction.
>
>
> - _From the empirical results, the authors seem to conclude that on OOD data, the models behave somewhat like a 1-nearest-neighbor classifier. The NCG accuracies are better than random but the raw NGC accuracies are often much too less to say that the model behaves like the 1-nearest-neighbour classifier unless I misunderstand the discussion. Furthermore, the improvement of robust training over natural training (in terms of NCG accuracy) is statistically significant, but the actual increases are still fairly small -- this might imply that the global behaviour is changing but not that drastically since the NCG accuracies of both the natural and robust training would be considered quite low. The results in Figure 6 seem to convey a similar message that the models continue to remain locally smooth -- the NCG accuracy drops as the distance increases. So there is some global effect but it does not appear to be significant._
>
> High NCG accuracy can only say that the model behaves like 1-nearest-neighbor (1NN) classifier on the support of the OOD inputs. It does not say that overall the model behaves like 1NN. What the statistical testing results in Section 3 shows is that the neural network has a prediction pattern that is more like a 1-nearest-neighbor than a random classifier. Based on the numbers we reported, we do agree that there are cases where the NCG behavior is not drastic. We do also agree that Figure 6 suggests that NCG is still kind of local, but compared to adversarial robustness, it has a much wider effect.
>
> Doesn't the difference between pixel space and feature space results boil down to whether we are enforcing smoothness across the end-to-end model or just across the final few layers of the model? In the latter case, it is not surprising that robust training is close to natural since the smoothness enforcement is only on part of the model and hence the effect is somewhat limited.
> In terms of the training process, yes. However, we still would like to point out that even if the magnitude of the difference between natural and robust training is smaller in the feature space, the difference there is still statistically significant.

---

> > ### Comment · Reviewer_Svr8 · 2022-12-13
> > **Regarding correlation between test accuracy and NCG accuracy**
> >
> > > We would like to clarify that we did not claim that higher NCG accuracy "implies" higher test accuracy. Instead, we claim that we found a positive correlation between NCG accuracy and test accuracy on multiple datasets. This result implies that the inductive bias that allows a model to have higher NCG accuracy may overlap with the inductive bias that is required to have high test accuracy on corrupted examples.
> >
> > My choice of the word "imply" was inaccurate. What I meant is the following:
> >
> > - If there is be a positive correlation between test accuracies and NCG accuracies, we would often simultaneously observe a high NCG accuracy and a high test accuracy (on corrupted data).
> >
> > - This means that, for most examples $x$ (with corrupted version $\tilde{x} = x + \Delta$ for corruption $\Delta$) in the test set, the label $\hat{y}(\tilde{x})$ predicted by the model matches the true label $y^\star(x)$ **and** the 1NN label $y_{\text{NN}}(\tilde{x})$, which would mean that the true label $y^\star(x)$ and the 1NN label $y_{\text{NN}}(\tilde{x})$ are same most of the time.
> >
> > - First, this implicit implication that the true label $y^\star(x)$ and the 1NN label $y_{\text{NN}}(\tilde{x})$ match seems restrictive. Am I misunderstanding something in this setup?
> >
> > - Moreover, this also seems to implicitly mean that the corruption added to the original image (whose label is probably considered the true label) does not change the 1NN (in the $\ell_2$ distance in pixel space) label of the image. This means that the corruption is added is a specific way that ensures that the 1NN label $y_{\text{NN}}(\tilde{x})$ of the corrupted example $\tilde{x}$ does not change.
> >
> > - If the corruption $\Delta$ is large enough to change the 1NN label (true labels are inherent and don't change), that is $y^\star(x) \not= y_{\text{NN}}(\tilde{x})$, then for any one model, either NCG accuracy would be high or test accuracy would be high, but not both since the same predicted label $\hat{y}(\tilde{x})$ cannot match the true label and the 1NN label simultaneously since the corruption ensured that the 1NN label has changed.
> >
> > What am I misunderstanding? If my understanding is correct, it seems that the correlation between the test accuracy and the NCG accuracy could be a byproduct of the implicit properties of the corruption mechanism rather than an effect of the model inductive bias.

---

> > > ### Author Response · Authors · 2022-12-19
> > > **Thank you for responding**
> > >
> > > > What am I misunderstanding? If my understanding is correct, it seems that the correlation between the test accuracy and the NCG accuracy could be a byproduct of the implicit properties of the corruption mechanism rather than an effect of the model inductive bias.
> > >
> > > Sorry for not being clear enough in the first response. We would like to make the second bullet point a bit more clear. Recall the result in Section 4.1 (this should be the result you were referring to in the initial comment), we claim that NCG correct examples are statistically more likely to be correctly classified. This means that
> > > conditioning on $y_{NN}(\tilde{x}) = f(\tilde{x})$, it is more likely for $f(\tilde{x}) = y*(x)$ then in the case where $y_{NN}(\tilde{x}) \neq f(\tilde{x})$. Thus, it is not exactly the way being described in the second bullet point.
> > >
> > > On the second and fourth bullet points, we would like to note that when computing the $y_{NN}(\tilde{x})$, we only search in training set for the 1NN label. Given $x$ is a test example and $\tilde{x}$ is the corrupted version of the test example, it is possible that $y*(x) \neq y_{NN}(x)$ (when $\delta$ is zero). Therefore, $y*(x) = y_{NN}(\tilde{x})$ may not happen most of the time. For instance, in CIFAR10-C, only $27.74$% of the corrupted examples across all corrupted data satisfy $y*(x) = y_{NN}(\tilde{x})$.
> > >
> > > We understand your concern that the corruption mechanism can be a confounding factor in this correlation between NCG accuracy and test accuracy on corrupted data (Section 4.1 result). However, in Table 4, our claim is statistically significant most of the time across different corruption types. This means that at least on the 18/15 corruption types we tested, these corruption mechanisms are unlikely to intervene with our claim.
> > >
> > > For the third bullet point, we were wondering what it means to be restrictive. Could you please elaborate more on this?

---

> ### Author Response · Authors · 2022-12-10
> **Response (3/4)**
>
>
> - _Figure 1 is not very clear. How is it related to OOD generalization? The smoothness of the model might be different in different directions (especially in high dimensions), but how does that connect with OOD generalization? The learned model has increased or decreased smoothness is different direction based on the decision boundary it is learning. Moreover, It might be useful to consider the directions where the smoothness increases after robust training (relative to natural training)._
>
> In Figure 1, there are two kinds of smoothness presented. One is the smoothness given by adversarial training, where smoothness is enforced to all directions in a norm ball with radius $r$. The other kind of smoothness is more of what NCG is measuring, where its smoothness extends towards a certain direction more (the line labeled with OOD distance), and that direction points more towards some natural OOD image.
>
> It is known that adversarial training improves the first kind of smoothness (through measuring adversarial robustness). In this work, we show that adversarial training also improves the second kind of generalization. The increase in the NCG accuracy after robust training implies that the models extend their decision boundary towards certain natural images more. OOD generalization means the way that neural networks respond to OOD inputs. As the second kind of smoothness can extend to the region where OOD data lies, this can help us understand OOD generalization.
>
> - _Why should we focus on the 1-nearest-neighbour classifier with  distance in pixel/feature space as the base model?_
>
> For the choice of the distance measure, there isn’t a consensus on what’s the best distance for images. Therefore we pick \ell_2 distance in the pixel and feature spaces as they are two of the most straightforward options.
>
> - _For the choice of the classifier to use as a probe for the decision boundary of a neural network, we want to choose a common classifier that is easy to reason about so that when the decision boundary of the network matches the classifier well, we know its meaning. This leaves us just a handful of options – linear classifiers, decision trees, and nearest neighbors. Among them, the linear classifier and decision tree requires each dimension of the feature to be meaningful, which is not the case for image data as interpreting each pixel independently contains too little information and each feature in the feature space doesn't have a concrete meaning. This makes 1 nearest neighbor classifier the most natural choice._
>
> We agree that there are other kinds of models and distance measures that are worth exploring. As this study established the experimental setup and method to study a model’s decision boundary on OOD data, researchers can easily adapt other models or measures to our setup.
>
>
> - _Where is figure 4 referenced? It seems to appear out of the blue and not discussed anywhere in the main paper. However, I might have missed something in the text._
>
> Thanks for the reminder. We have mistakenly left the figure from the previous version in the paper. We have removed it from the latest revision.
>
> - _The theorem statement appears incomplete with respect to the role of ϵ and C beyond just universal constants -- the sample complexity deteriorates with smaller  but it is not clear what tradeoff is present and why one should not just pick ϵ≈1/2 and get a better sample complexity than that of O(Clog⁡C). I understand that ϵ pertains to the distribution of the data. But the role of ϵ needs to be presented rigorously in the theorem statement._
>
> The point is that we choose the distribution based on epsilon, as a way to increase the separation between the two sample complexities. If you want a large separation, then you set epsilon to be very small. Then you construct the distribution based on epsilon.

---

> ### Author Response · Authors · 2022-12-10
> **Response (4/4)**
>
>
> - _Regarding "the behavior on OOD data is far from random", why should we expect the OOD behavior of any trained model to be random?_
>
> What we mean is that before this work, there was a lack of knowledge on how a model would behave on OOD inputs. Specifically, we are not aware of any model for the distribution of the network’s prediction on OOD data. Our work highlights that it is not uniformly random over the class labels. Using NCG accuracy (1-NN agreement) is how we probe the network’s behavior, and we believe this is a first step to better understand OOD generalization ([1]). There is also a lot of work on the bias of neural networks on OOD data, such as exemplar-based generalization ([2]). This other area is also motivated by similar questions as our work. For example, understanding zero-shot learning, spurious correlations, representations, etc. However, a main difference is that we explicitly focus on OOD data (held-out classes and corruptions).
>
> [1] Jiang, Yiding, et al. "Methods and analysis of the first competition in predicting generalization of deep learning." NeurIPS 2020 Competition and Demonstration Track. PMLR, 2021.
>
> [2] Dasgupta, Ishita, Erin Grant, and Tom Griffiths. "Distinguishing rule and exemplar-based generalization in learning systems." International Conference on Machine Learning. PMLR, 2022.
>
> - _Why should one consider corrupted data (used in the experiments) OOD especially if the corruption does not change the class of the nearest-neighbor (in pixel/feature space)?_
>
> Corrupted data is considered OOD because they are considered sampled from a different distribution during testing (these corruptions are not applied during training).
>
>
> - _Are all global effects of local smoothing positive or neutral or can there be negative global effects?_
>
> No. We believe that there can be negative effects. There is a work ([1]) pointing out that excessive local smoothing can produce negative effects.
>
> [1] Jacobsen, Jörn-Henrik, et al. "Exploiting excessive invariance caused by norm-bounded adversarial robustness." arXiv preprint arXiv:1903.10484 (2019).

---

### Author Response · Authors · 2022-12-10
**General comments**

We thank all the reviewers for their insightful and constructive feedback. Based on the comments, we have made some revisions to the paper. The new contents are highlighted in a different color. The key messages of this work are:
- We uncover this nearest category generalization (NCG) behavior for neural networks on out-of-distribution (OOD) data, where OOD data tends to be predicted as the label of the nearest training example.
- We look into this NCG property and find that increasing local smoothness increases this property, and this property seems to be depending more on the neural network’s inductive bias rather than the loss function.
- Our result suggests that the NCG property can potentially be used in the case where OOD detection cannot be done well. When an OOD example cannot be detected, the NCG property can be used to reason how some predictions are given.

One common concern from reviewers is that they are unclear about the motivation of NCG. To clarify, the study of NCG begins with the fact that how neural networks perform on OOD examples is still an open question. The main purpose of this study is to use NCG as a measurement to probe the decision boundary of a neural network on the support of OOD data.

Another common concern is that reviewers think the connection between OOD detection and NCG can be strengthened. In Section 5.1, we have shown that there are cases where OOD detection is extremely sample inefficient. In the case where an OOD detector misclassifies an OOD example as in-distribution, the model will have to give a prediction on examples that it does not expect. Therefore, it is important to understand what kind of tendency the model has on OOD examples, especially on OOD examples that are misclassified as in-distribution examples.

As a concrete example, we train a model on MNIST images of 0-8 and use the model for prediction on images of 9s. We also train an OOD detector – ODIN [1] – which has a .951 true positive rate and .875 false negative rate. In this example, many images of 9s cannot be easily picked out by OOD detectors and will be treated as in-distribution examples. Thus, it is important to know what kind of prediction will be given to these 9s. As a result, many 9s are predicted as 4s, and this can be explained by nearest category generalization. We have added these discussions in Section 5.

[1] Liang, Shiyu, Yixuan Li, and Rayadurgam Srikant. "Enhancing the reliability of out-of-distribution image detection in neural networks." arXiv preprint arXiv:1706.02690 (2017).

---

### Decision · Action_Editors · 2023-01-09

**Recommendation:** Accept with minor revision

**Comment:**

The authors present an interesting metric to measure the OOD behaviour of models by measuring a surrogate for the smoothness of the model. This metric is evaluated for various different training schemes and different forms of OOD samples, highlight some insights, such as (i) adversarially trained models have increased smoothness relative to naturally trained models far beyond the robust radius, and (ii) models with higher smoothness tend to have higher testing accuracy of OOD data generated via corruption. However, there are some following issues to be addressed carefully, based on the valuable comments from three qualified reviewers.

1) According to Reviewer Svr8, the metric and it's properties need improved motivation and explanation and many things are unclear to a reader (see our reviews and discussions). Furthermore, various interpretations of empirical results do not appear to be well supported by the actual results, necessitating improved presentation of the results.

2) According to Reviewer TdWQ,
2.1) "NCG accuracy provides a new way to evaluate training methods on unlabeled data that cannot be labeled using other information": If "evaluate" means to measure some goodness, this might be an overstatement.
2.2) "This indicates that having higher NCG accuracy may be a desirable property, as it can encourage better predictions on corrupted data.": This sentence sounds like the causal statement that "making NCG accuracy higher leads to better predictions on corrupted data." This is not confirmed by the presented experiments. Consider adding experiments or adjusting the statement.
2.3) "Nonetheless, they both suffice to provide insight into the OOD prediction behavior of robust and normal networks." This sounds as if there were nothing further to be studied.
2.4) Consider rephrasing "accuracy" in the term "NCG accuracy". Accuracy sounds like a goodness measure of performance on a task that we wish to solve.
2.5) "NCG can be much easier" than OOD detection: This can mean the OOD detection problem cannot be reduced to NCG. If so, that would be a contradiction. Please consider adding comments on that.

3) According to Reviewer eC2G, the manuscript does not fully support the validity of NCG accuracy. The authors claim "robust networks are more likely to classify OOD data with the class label of the nearest training input", which seems a reasonable claim given the experimental results. However, robust networks could classify in-distribution data with the class label of the nearest training input as well. If this happened, NCG accuracy would not be sufficiently informative to probe OOD data. I would like to see whether this actually happens or not if possible.

Hence I recommend **major** revisions, altough TMLR systems can only show **minor** revisions. We do hope that the authors can carefully respond the above comments one by one and upload a updated version for the further check.


**Audience:**

Yes

**Claims And Evidence:**

Yes

---

> ### Author Response · Authors · 2023-03-02
> **Response to revision comments**
>
> We have uploaded the camera ready version and addressed the comments in the revision. The following are the responses to each comments.
>
> `1) According to Reviewer Svr8, the metric and it's properties need improved motivation and explanation and many things are unclear to a reader (see our reviews and discussions). Furthermore, various interpretations of empirical results do not appear to be well supported by the actual results, necessitating improved presentation of the results.`
>
> We have added more discussion on the exemplar-based generalization to motivate NCG in the new Introduction section.
>
> `2.1) According to Reviewer TdWQ, 2.1) "NCG accuracy provides a new way to evaluate training methods on unlabeled data that cannot be labeled using other information": If "evaluate" means to measure some goodness, this might be an overstatement.`
>
> We tone down this sentence with "understand a model's prediction" instead of "evaluate" (in Section 1.1).
>
> `2.2) "This indicates that having higher NCG accuracy may be a desirable property, as it can encourage better predictions on corrupted data.": This sentence sounds like the causal statement that "making NCG accuracy higher leads to better predictions on corrupted data." This is not confirmed by the presented experiments. Consider adding experiments or adjusting the statement.`
>
> We replace "it can encourage better predictions" with "it is positively correlated with better predictions", which is a more accurate statement.
>
> `2.3) "Nonetheless, they both suffice to provide insight into the OOD prediction behavior of robust and normal networks." This sounds as if there were nothing further to be studied.`
>
> We have revised it as "Nonetheless, they both {\color{ForestGreen} can be used by future work} to provide insight into the OOD prediction behavior of robust and normal networks."
>
> `2.4) Consider rephrasing "accuracy" in the term "NCG accuracy". Accuracy sounds like a goodness measure of performance on a task that we wish to solve.`
>
> We replace all NCG accuracy as NCG score.
>
> `2.5) "NCG can be much easier" than OOD detection: This can mean the OOD detection problem cannot be reduced to NCG. If so, that would be a contradiction. Please consider adding comments on that.`
>
> We rephrase this sentence as "our result suggests that achieving a high NCG score can be much easier than achieving a good detection rate in some cases".
>
> `3) According to Reviewer eC2G, the manuscript does not fully support the validity of NCG accuracy. The authors claim "robust networks are more likely to classify OOD data with the class label of the nearest training input", which seems a reasonable claim given the experimental results. However, robust networks could classify in-distribution data with the class label of the nearest training input as well. If this happened, NCG accuracy would not be sufficiently informative to probe OOD data. I would like to see whether this actually happens or not if possible.`
>
> We have added an experiment for this in Appendix D.5. In short, robust networks do have a slightly higher NCG score on in-distribution data than naturally train networks. However, this increase in NCG score from natural to robust training is smaller for in-distribution data than for OOD data. This means that our claim not only holds for OOD data, but also holds for in-distribution data. However, the phenomenon of NCG is more prominent on OOD data.